# Exploring the potential of Direct Feedback Alignment for Continual Learning

**Sara Folchini**                                                           *sara.folchini@gmail.com*
*International Institute for Advanced Studies (SISSA)*
*Trieste, Italy*

**Viplove Arora**                                                          *viplovearora92@gmail.com*
*International Institute for Advanced Studies (SISSA)*
*Trieste, Italy*

**Sebastian Goldt**                                                              *sgoldt@sissa.it*
*International Institute for Advanced Studies (SISSA)*
*Trieste, Italy*

**Reviewed on OpenReview:** *https://openreview.net/forum?id=MRZQrn7JEG&noteId=bzHvuL5L4d*

## Abstract

Real-world applications of machine learning require robustness to shifts in the data distribution over time. A critical limitation of standard artificial neural networks trained with backpropagation (BP) is their susceptibility to catastrophic forgetting: they "forget" prior knowledge when trained on a new task, while biological neural networks tend to be more robust to catastrophic forgetting. While various algorithmic ways of mitigating catastrophic forgetting have been proposed, developing an optimization algorithm that is capable of learning continuously remains an open problem. Motivated by recent theoretical results, here we explore whether a biologically inspired learning algorithm like Direct Feedback Alignment (DFA) can mitigate catastrophic forgetting in artificial neural networks. We train fully-connected networks on several continual learning benchmarks using DFA and compare its performance to vanilla backpropagation, random features, and other continual learning algorithms. We find that an inherent bias of DFA, called "degeneracy breaking", leads to low average forgetting on common continual learning benchmarks when using DFA in the Domain-Incremental and the Task-Incremental learning scenarios. We show how to control the trade-off between learning and forgetting with DFA, and relate different modes of using DFA to other methods in the field.

## 1 Introduction

Neural networks have demonstrated remarkable achievements in various domains, including image recognition (Krizhevsky et al., 2012; LeCun et al., 2015; Simonyan & Zisserman, 2015; He et al., 2016; Dosovitskiy et al., 2021) and natural language processing (Devlin et al., 2019; Howard & Ruder, 2018; Radford et al., 2018; Brown et al., 2020; OpenAI, 2024). However, a significant drawback of traditional neural networks is their susceptibility to catastrophic forgetting (Goodfellow et al., 2015; McCloskey & Cohen, 1989). Catastrophic forgetting refers to a loss in performance on previously learned tasks when the network is trained on new tasks or data distributions. This limitation impedes the ability of neural networks to continuously learn and adapt to evolving environments, limiting their practical applicability in real-world scenarios.

Continual learning (CL) focuses on developing algorithms and techniques that enable systems to maintain old knowledge, while also being able to learn new information. This requires resolving the "stability–plasticity dilemma" (Carpenter, 1986; Mermillod et al., 2013): a model needs plasticity to obtain new knowledge and adapt to new environments, while also requiring stability to prevent forgetting of previous information.

Typical approaches to CL include dynamic architectures (Razavian et al., 2014), progressive learning (Fayek et al., 2020), regularisation techniques (Kirkpatrick et al., 2017; Zenke et al., 2017), or episodic memory replay (Lopez-Paz & Ranzato, 2017; Chaudhry et al., 2019; Rebuffi et al., 2017; Shin et al., 2017; Kamra et al., 2017; Seff et al., 2017); see (De Lange et al., 2021) for a review.

Meanwhile, biological neural networks do not suffer from catastrophic forgetting nearly as badly as artificial neural networks. Consequently, several approaches to mitigate catastrophic forgetting have taken direct inspiration from biology: for example, the complexity of synaptic plasticity in biological neurons inspired regularisation approaches such as Elastic Weight Consolidation (EWC) (Kirkpatrick et al., 2017) and Synaptic Intelligence (Zenke et al., 2017).

Here, we continue this line of thought by investigating the potential of a biologically plausible learning algorithm to mitigate catastrophic forgetting: Direct Feedback Alignment (DFA). Proposed by Nøkland (2016) as a biologically plausible algorithm to train neural networks, DFA propagates the error signal through fixed, random feedback connections directly to the hidden layers, thereby solving the weight transport problem that plagues vanilla backpropagation (Grossberg, 1987; Crick, 1989). Decoupling the weight updates of different layers enables parallel and local updates, which are considered key features of synaptic updates in the brain (Bengio et al., 2015).

We focus on the potential of DFA not just because of its greater biological plausibility compared to vanilla backpropagation. We are encouraged by the recent analysis of Refinetti et al. (2021), who showed that DFA has a *degeneracy breaking* property. To learn with DFA, the neural network has to first align its weights (to some extent) with the feedback matrices to ensure that the error signal can be backpropagated efficiently (Lillicrap et al., 2016). This alignment has the effect that neural networks trained by DFA always converge to the same region in the loss landscape, independently of their initialisation, and in contrast to networks trained by vanilla backpropagation. In other words, DFA will drive a neural network to a specific region in the loss landscape which depends on the feedback matrices, thereby breaking the degeneracy of the solutions of vanilla SGD. In this work, we ask whether we can use the influence of the DFA feedback matrix on the weights learnt by a neural network to mitigate catastrophic forgetting. We formulate and test two different hypotheses for how DFA can facilitate continual learning.

Our **first hypothesis** is that using the *same* feedback matrix for different tasks in a continual learning curriculum prevents catastrophic forgetting by implicitly biasing the weights to a single region of loss landscape, as they always need to align with the same feedback matrix. In this case, DFA would act as a an *implicit regulariser*, similar to other algorithms like Elastic Weight Consolidation (EWC) (Kirkpatrick et al., 2017), a popular implicit regularisation technique. We will call this approach **DFA-same**.

Our **second hypothesis** is that using *different* feedback matrices for each task will effectively update the weight matrices in different directions for each task, thus preventing catastrophic forgetting. This **DFA-diff** approach is inspired by various CL algorithms that explicitly orthogonalise gradients (Zeng et al., 2019; He & Jaeger, 2018; Bennani & Sugiyama, 2020). This idea can also be motivated from a neuroscientific perspective given recent evidence that neural population codes orthogonalize with learning (Flesch et al., 2022; Failor et al., 2021; Zeng et al., 2019).

Prior work has also explored DFA's application in online learning contexts, where the primary goal is efficient adaptation to nonstationary data streams rather than addressing catastrophic forgetting (Lindsey & Litwin-Kumar, 2020).

Since efficient training of convolutional neural networks (CNNs) with DFA remains an open problem (Crafton et al., 2019; Launay et al., 2020), we focus on fully-connected networks and limit ourselves to simple benchmark datasets based on MNIST, FMNIST where these architectures achieve good performance. We use fixed pre-trained convolutional layers before the trained fully-connected network for CIRFAR10. In particular, we use the split and the permuted manipulations of MNIST, FMNIST that turn the 10-class classification tasks into a set of successive classification tasks (Kirkpatrick et al., 2017; Zenke et al., 2017). For CIFAR10, we limit ourselves to the split manipulation.

The contributions of this paper are threefold:

1. We empirically show that DFA is competitive at Continual Learning to vanilla back-propagation and other baselines, such as random features (RF) and Elastic Weight Consolidation (EWC).

2. We benchmark the performance of different DFA strategies for Continual Learning, either sharing or changing feedback matrices between tasks, and show how the best strategy depends on the type of continual learning task and the network architecture.

3. We show that the scale of the feedback matrix allows one to trade-off plasticity vs. stability in continual learning, beyond what can be achieved by choosing layer-specific learning rates.

Exploring the potential of DFA to mitigate catastrophic forgetting both deepens our understanding of the working mechanism behind DFA by testing it in a continual learning setting, and it adds to the tool set that can be tested on continual learning problems. The remainder of this paper is organized as follows: In section 2, we provide the details of the experimental setup. Section 3 presents our results by benchmarking DFA on various continual learning tasks and contrasting its performance with various baselines. Section 4 is dedicated to a discussion of our results and gives some concluding perspectives.

## 2 Methods

**Direct Feedback Alignment (DFA)**  DFA presents a departure from the traditional backpropagation algorithm (Rumelhart et al., 1988), which relies on weight symmetry and precise weight updates for accurate error propagation. Instead, DFA employs random feedback matrices that are decoupled from the forward weight matrices, enabling more flexible weight updates without the constraints imposed by weight symmetry (Nøkland, 2016).

To state the DFA weight updates clearly, and to contrast them with vanilla backpropagation, we consider mini-batches of input-output pairs $(x, y)$ that we want the network to learn. For simplicity, we consider a simple network composed of two fully-connected layers and one softmax layer at the end. We denote as $W_i$, $h_i$ and $a_i$ the weight matrix, activation function and activations of layer $i$; The output of the network is $\hat{y}$ and the activations of the output layer are $a_y$. The error $e$ is the derivative of the loss $J$ and in case of a cross-entropy loss it is equal to:

$$e = \delta a_y = \frac{\partial J}{\partial a_y} = \hat{y} - y \tag{1}$$

Denoting $\odot$ as the Hadamard product, for BP (on the left) and DFA (on the right), the gradients of the hidden layers are calculated as

$$\text{BP} \qquad\qquad\qquad\qquad \text{DFA}$$

$$\delta W_3 = -eh_2^T \qquad\qquad\qquad\qquad \delta W_3 = -eh_2^T \tag{2}$$

$$\delta W_2 = \frac{\partial J}{\partial a_2} = (W_3^T e) \odot f'(a_2) h_1^T \qquad\qquad \delta W_2 = \frac{\partial J}{\partial a_1} = (B_2 e) \odot f'(a_2) h_1^T \tag{3}$$

$$\delta W_1 = (W_2^T \delta a_2) \odot f'(a_1) x^T \qquad\qquad \delta W_1 = (B_1 e) \odot f'(a_1) x^T \tag{4}$$

While the final layer (the readout layer) is updated in the same way with both algorithms, the weight updates for the other layers are all different. BP implements the exact gradient for each layer by applying the chain rule to compute the derivatives of the loss; instead, DFA keeps only the error term and substitutes the derivative of the following layer by an entry of the feedback matrix. A detailed theoretical analysis of the evolution of the update dynamics can be found in Refinetti et al. (2021).

In the case of DFA-same, the feedback matrices ($B_1$ and $B_2$) are initialized according to a uniform distribution and kept unchanged during the training of all tasks. According to the degeneracy breaking of DFA discussed in the introduction, DFA-same could alleviate catastrophic forgetting by keeping the weights of the network when training on the second task close to the weights learnt on the first task, etc.

In DFA-diff, we use a different feedback matrix for each task, by sampling the new ones from the same Uniform distribution used to sample the first one. The new feedback matrix is used in the backward pass to map the error to the activations of each layer. Since we use the same hidden size for every layer, the dimension of this matrix is output size x hidden size x number of layers. Our hypothesis is that the different feedback matrices may bias the dynamics of the networks in different directions of the weight landscape for every task. In fact, due to the large dimensions of the matrix and their sampling from uniform distributions, the new feedback matrices are approximately orthogonal to the previous ones. The gradient alignment will then be directed to approximately orthogonal directions for every task. Making the gradients explicitly orthogonal to each other is known to help against catastrophic forgetting, and the idea is exploited in methods such as Orthogonal Weight Modification (OWM) (Zeng et al., 2019), conceptor-aided backprop (He & Jaeger, 2018), and orthogonal gradient descent (Bennani & Sugiyama, 2020), because the features of different tasks are learned along orthogonal manifolds, and the weights updates do not interfere with the previous ones. Appendix A confirms that the gradients are indeed orthogonal when two networks are trained on the same dataset with orthogonal feedback matrices.

**Benchmark datasets**  We report results on the FashionMNIST (FMNIST) dataset (Xiao et al., 2017), CIFAR10 (Krizhevsky, 2009) dataset, and the MNIST dataset (Deng, 2012). We do not apply data augmentation techniques, except for CIFAR10, where we pre-process the images through a pre-trained VGG11 convolutional network. These convolutional layers are pre-trained on Imagenet; this process does not extract any feature specific for CIFAR10, as we show in Appendix B.

The CL benchmark tasks we use are: In our experimental evaluations, we use:

1. Permuted FMNIST (pFMNIST), where we generate a sequence of learning tasks by permuting the pixels of each image; This is a task that requires learning without relying on the previous task's features, like edges or corners. However, from the technical point of view, the features are similar to each other (Kemker et al., 2017) (See higher similarity in Appendix C, panel B). This procedure is useful as a baseline to the combination of DFA with convolutional pre-trained layers, in which useful features are already extracted ((Crafton et al., 2019)).

2. Split FMNIST (sFMNIST) and split CIFAR10, where we split the original dataset of 10 classes into five smaller datasets with two disjoint classes for each. The resulting smaller datasets will have very different statistical characteristics, so a model trained sequentially on them needs to be able to incrementally learn new information with dramatically different feature representations.

We show an example of the split and permuted FMNIST in Figure 1, panel A and the comparison of the class-wise similarity of the split and permuted datasets in Panel B (See Appendix C for a description of the similarity measures used).

**Experimental setup**  We test DFA in three continual learning scenarios (van de Ven & Tolias, 2019) (Panel C, fig. 1):

1. In the Domain-IL scenario, the network does not have the information about the task-ID during inference. The whole architecture is shared among the tasks, including the output layer.

2. In the Task-IL scenario, the network can use the task-ID during both training and testing. We implement this approach by having task-specific output layers that are updated only during the training on the corresponding task and used whenever testing on the same task.

3. In the Class-IL scenario the network can use the task-ID only during training; while it has to infer the task-ID of the test images. We implement this approach by having task-specific output layers that are updated only during the training on the corresponding task (to mimic a masked cross entropy loss) and used an aggregation of all the output heads for prediction.

We show in Figure 4 and in Appendix A that the weight alignment and gradient alignment of DFA are preserved if the output layer is re-initialized. These tests are important because the degeneracy breaking

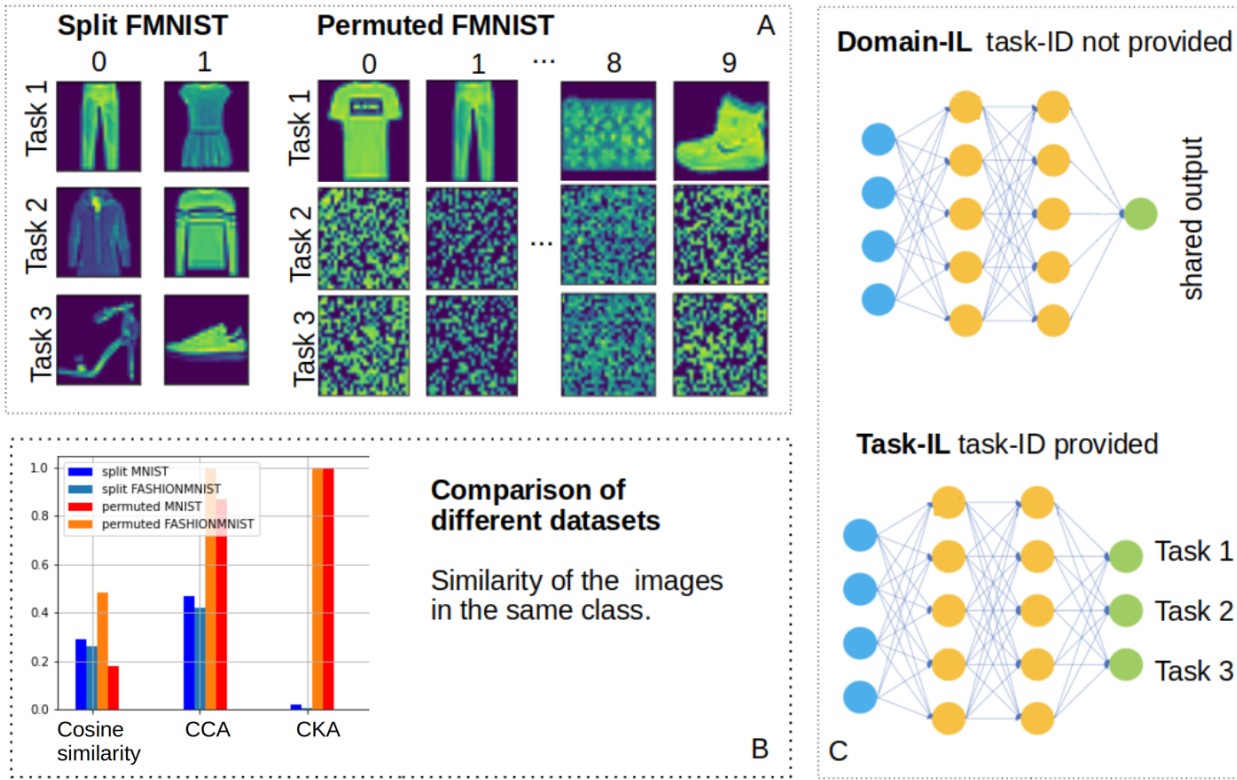

Figure 1: **A** One example of the first three tasks in the split (Binary classification) and permuted dataset (classification). **B** Similarity measures of the images in the same class averaged over the different classes of the Split and Permuted datasets. The permuted-FMNIST dataset is the one where the images have the highest similarity among tasks, while split-FMNIST has very different images in the same class. **C** One example of a three-layer architecture used in the Domain-IL and Task-IL scenarios. In the second case, one output node is dedicated to training and evaluating one specific task.

feature of DFA was previously observed in networks with weights sampled from independent distributions, while in this case only the output layers of the networks in comparison are different.

In our experiments, we use 3-layer Fully-Connected Networks with 1000 neurons in each hidden layer. We train the networks for a maximum of 1000 epochs (an impact of this choice is expanded in F) and apply early-stopping by halting the training as soon as the network overcomes 99% training accuracy. All layers are initialized using the Xavier uniform initialization (Glorot & Bengio, 2010). We choose a logistic activation function in the output layer and ReLU in the other layers. The loss function is cross-entropy.

**Evaluation metrics**  Performance in continual learning is difficult to report by one single measure due to the stability–plasticity trade-off. We will summarize the performances of the algorithms under study using average performance (AP) and average forgetting (AF) (Lopez-Paz & Ranzato, 2022; Chaudhry et al., 2018), which we visualize in fig. 2. To define average forgetting and average performance, we consider a number of tasks $T$ and denote by $\text{Acc}_{ij}$ the accuracy of the network on the $j$th task at the end of training on the $i$th task, where $j, i \in \{1, \ldots, T\}$. Average performance quantifies the performance of the network by calculating the mean test accuracy of the network on each task, directly after training on it:

$$\text{AP} = \frac{1}{T} \sum_{i=1}^{T} \text{Acc}_{i,i}. \tag{5}$$

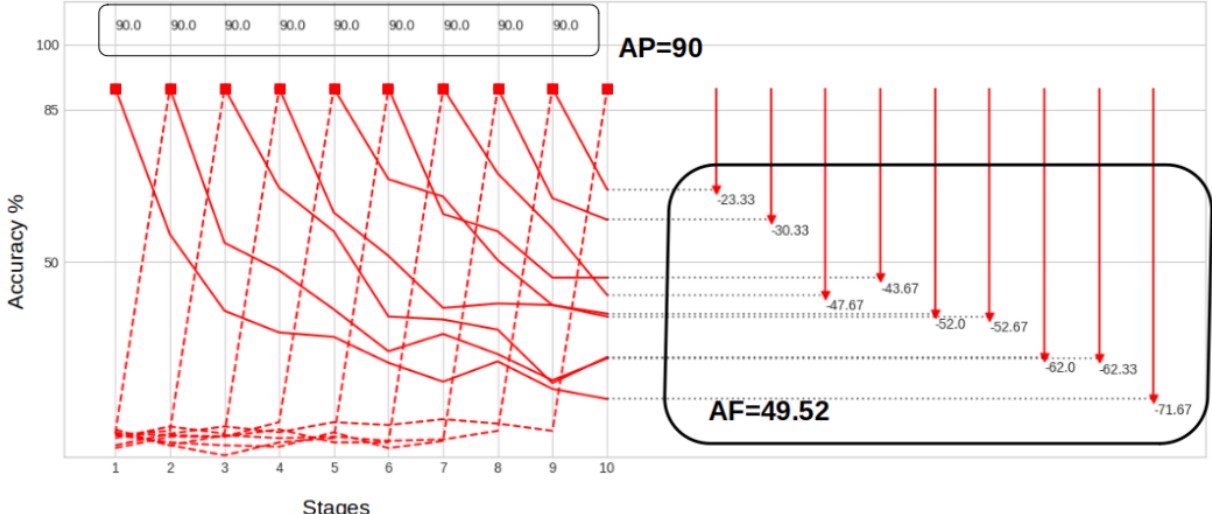

Figure 2: Illustration on Average Forgetting (AF) and Average Performance (AP) in the case of back-propagation on the p-FMNIST task in the Task-IL scenario, where the network has a separate head for each task. Average performance describes the test accuracy of the network, averaged over the tasks, while average forgetting quantifies the average of the difference in accuracy on a given task after training on it and at the end of the whole training trajectory.

Average forgetting measures how much the model loses accuracy on previous tasks during continual learning:

$$\text{AF} = \frac{1}{T-1} \sum_{i=1}^{T-1} \text{Acc}_{i,i} - \text{Acc}_{i,T} \tag{6}$$

Additionally, we measure Forward transfer as defined in Chen et al. (2023) in Appendix D.

**Baselines and hyper-parameters tuning**  To benchmark the performance of DFA, we consider the following additional baselines:

1. Backpropagation is the most important baseline, also known as fine-tuning: we train the same fully-connected networks without DFA and, when it is convenient as discussed in Section 3.4, a Dropout layer (Srivastava et al., 2014) after each layer, which we find helps with generalization, in accordance to the literature (Mirzadeh et al., 2020b) Mirzadeh et al. (2020a) (0.2% in the first layer and 0.5% in the other layers, excluding the output layer). In the BP-ablated version, dropout is not used. We perform a grid-search optimization for the learning rate in the range between 1e-2 and 1e-4.

2. Random Features (Rahimi & Recht, 2008; 2009) are fully-connected networks in which we train only the readout layer with BP. The first and the intermediate layers are kept unchanged during the training phase. This model can be applied only in the Task-IL scenario because it requires a different output layer for each task in the testing phase. The motivation for using this model is to understand the performance of a neural network that does *not* learn data-dependent features, akin to the lazy regime (Chizat et al., 2019). We use a learning rate of 1e-2.

3. Elastic weight consolidation (EWC) (Kirkpatrick et al., 2017) adds a regularisation term on the loss that penalizes the change of the weights that are more important for previous tasks. It achieves this by pre-multiplying the BP weight updates using the inverse of the diagonal approximation of the Fisher information matrix of the model. In our EWC experiments, we chose a learning rate of 1e-3 and an "importance" of 1000; lambda is set to 0.4 by default, except for the Domain-IL experiment on the CIFAR10 experiment (architecture with pretrained fixed concolutional layers).

Table 1: Results on the FMNIST dataset in terms of Average Performance (AP) and Average Forgetting (AF) in the different datasets and scenarios (Task-IL and Domain-IL) for all the methods (DFA–diff, DFA–same, BP, BP–ablated, EWC, RF ). In the first column, the networks are optimized for maximum average performance and in the second column for minimum average forgetting (See section 2 for the definition of average performance and average forgetting). We present these results visually in Figure 5. In the case of minimized forgetting, we report here the minimum average performance % values above the RF baseline.

| | | AP | AF | AP | AF |
|---|---|---|---|---|---|
| | | Maximizing Accuracy | | Minimizing forgetting | |
| Split FMNIST | DFA-diff | 100 ±0 | 9.8±2 | 93.9 ±0.1 | 0 ±0 |
| Task-IL | DFA-same | 100 ±0 | 29.4 ±1.2 | 95.4 ±0 | 0 ±0 |
| | BP | 100 ±0 | 15.0 ±2.3 | 89.5 ±0.1 | 0.3±0.3 |
| | BP ablated | 100 ±0 | 9.7±1.6 | 94 ±0 | 0 ±0 |
| | EWC | 99.2 ±0.2 | 4.5±1.5 | 99.1 ±0.2 | 3.4±1.8 |
| | RF | 93.4 ±0.1 | 0 ±0 | 93.4 ±0.1 | 0 ±0 |
| Permuted FMNIST | DFA-diff | 88.5 ±0.1 | 44.5 ±2.4 | 83 ±0 | 0 ±0 |
| Task-IL | DFA-same | 88.2 ±0.1 | 50.1 ±2.5 | 83 ±0 | 0 ±0 |
| | BP | 90.0 ±0.1 | 49.5 ±2.5 | 83 ±0 | 11.8 ±0.8 |
| | BP ablated | 90 ±0.1 | 47 ±2 | 82.8 ±0.1 | 0 ±0 |
| | EWC | 88.7 ±0.1 | 46.5 ±2.7 | 83 ±0 | 8.7±1.2 |
| | RF | 80.9 ±0.1 | 0 ±0 | 80.9 ±0.1 | 0 ±0 |
| Split FMNIST | DFA-diff | 100 ±0 | 45.8 ±2.4 | 94.0 ±0 | 35.5 ±1.6 |
| Domain-IL | DFA-same | 100 ±0 | 35.5 ±0.2 | 98.2 ±0 | 32.1 ±1.6 |
| | BP | 100 ±0 | 37.8 ±0.4 | 94.5 ±0.1 | 35.8 ±0.6 |
| | BP ablated | 100 ±0 | 40.5 ±1.7 | 98.4 ±0.1 | 34.7 ±0.7 |
| | EWC | 100 ±0 | 46.9 ±4.8 | 98.7 ±0.4 | 40.4 ±3.1 |
| Permuted FMNIST | DFA-diff | 88.3 ±0 | 71.8 ±1.2 | 82.3 ±0.1 | 29.1 ±0.9 |
| Domain-IL | DFA-same | 88.2 ±0.1 | 45.8 ±1.9 | 85.8 ±0 | 26.2 ±1.9 |
| | BP | 89.7 ±0.1 | 57.6 ±1.1 | 85.4 ±0.1 | 23.1 ±1.5 |
| | BP ablated | 88.7 ±0.1 | 49.4 ±0.4 | 84 ±0.1 | 26.2 ±2.4 |
| | EWC | 80.7 ±0.2 | 55.7 ±2.5 | 80.7 ±0.2 | 55.7 ±2.5 |

Additional baselines such as OWM or CAB are not taken as a comparison because they suffer form excessively expensive computational requirements. The architecture (number of layers and hidden size) and the batch size are the same among all the methods compared. For DFA, we use a learning rate of 0.01 and a Feedback matrix variance optimized in the range between the orders of 1e-8 and 1.

The error bars in table 1 are computed over a repetition of the experiment with 5 random seeds.

## 3 Results

### 3.1 DFA between the two ends of the plasticity-stability trade-off

Catastrophic forgetting arises in Backpropagation (Rumelhart et al., 1988) due to the fact that the weights of the network are updated in the direction of the minimum of the loss, irrespective of the previous tasks. In this case the network is flexible to fit the task at hand, but forgetting is high. Random features are at the other extreme, since only the weights in the output layer are trained on the task it results in a more rigid

Table 2: Results for split-CIFAR10 obtained by training a 3-Layer FC network after VGG11 with fixed pre-trained layers (Pre-trained on Imagenet). The legitimacy for this choice and hyperparameters is the same as in B. The results are reported in terms of Average Performance (AP) and Average Forgetting (AF) as defined in 2. In the first column, the networks are optimized for maximum average performance, and in the second column for minimum average forgetting. Further results on CIFAR10 are in F.

| | | AP | AF | AP | AF |
|---|---|---|---|---|---|
| | | Maximizing Accuracy | | Minimizing forgetting | |
| Split CIFAR10 | DFA-diff | 89.1 ±0.1 | 2.9±0.4 | 82.3 ±0.1 | 1.4±0.4 |
| Fixed Conv + 3L FC | DFA-same | 88.7 ±0.2 | 25.9 ±0.6 | 80.5 ±0.1 | 4.5±0.3 |
| Task-IL | BP dropout | 89.4 ±0.3 | 7.3±0.4 | 88.3 ±0.1 | 5.0±0.2 |
| | BP | 89.0 ±0.1 | 22.8 ±4.6 | 88.3 ±0.1 | 5.2±0.1 |
| | EWC | 89.1 ±0.1 | 17.1 ±2.8 | 83.72±0.15 | 2.5±0.5 |
| | RF | 85.7 ±0.1 | 0 | 66.1 ±0.8 | 0 |
| Split CIFAR10 | DFA-diff | 89.1 ±0.2 | 30.2 ±0.4 | 80.2 ±0.2 | 9.7±0.3 |
| Fixed Conv. + 3L FC | DFA-same | 88.6 ±0.2 | 24.4 ±0.4 | 81.5 ±0.1 | 8.1±0.3 |
| Domain-IL | BP dropout | 89.6 ±0.2 | 26.7 ±0.3 | 88.6 ±0 | 25.9 ±0.1 |
| | BP | 89.6 ±0.1 | 26.9 ±0.3 | 88.5 ±0.1 | 25.0 ±0.1 |
| | EWC | 89.4 ±0.1 | 28.2 ±0.5 | 84.9 ±0.1 | 16.0 ±0.3 |

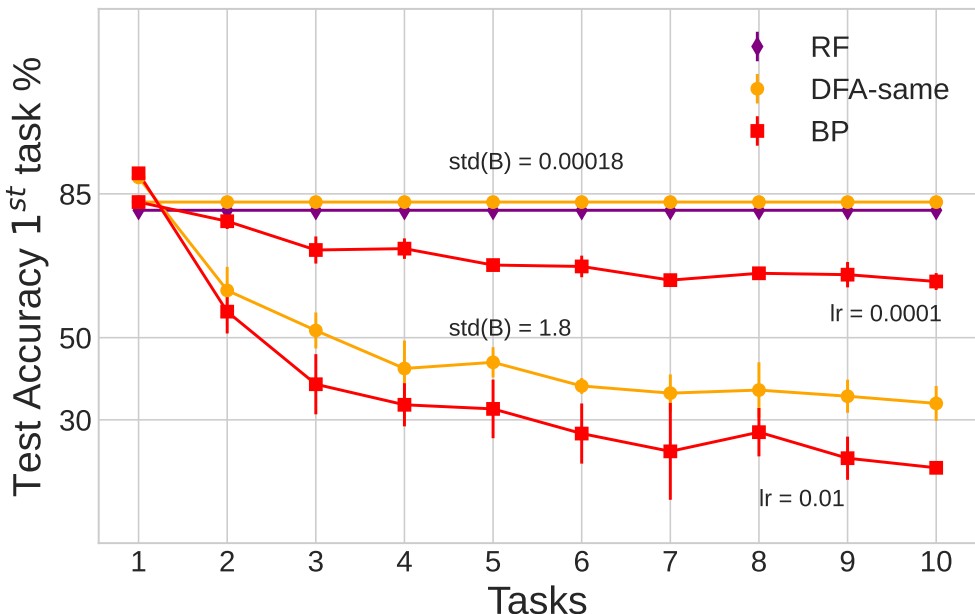

Figure 3: Performance of the first task throughout learning all tasks evaluated on the permuted-FMNIST (above) in the Task-IL scenario. DFA-same is more powerful than RF (purple line). Compared to BP, DFA-same has less forgetting in both cases: when we optimize the models for maximum performance and for least forgetting. The errorbars are the standard deviation over at least 3 runs. A complete report of test accuracy drop of the first task can be found in Appendix G.

method that retains previous performances. BP and random features therefore delineate the two extremes between which we would like to interpolate: we expect DFA to be less susceptible to catastrophic forgetting than BP, while being able to learn data-dependent features that allow it to beat the performance of random

features. We report the results of our experiments that we obtain by optimizing the hyper-parameters for maximum average performance or for minimum average forgetting in table 1 for FMNIST; in table 2 (after pre-trained convolutional layers) and Appendix F (directly on the images) for CIFAR10. Results for MNIST can be found in Appendix E.

When we train DFA-same and BP while optimizing the hyperparameters for accuracy, we can see that DFA-same can reach the same performance as BP of 100% average performance on the split FMNIST dataset (where the tasks are binary classifications). On the permuted FMNIST and split CIFAR10 datasets, there is minimal difference (less that 1.5%) between BP and DFA. Thus, DFA allows the network to adapt and learn new tasks efficiently. In terms of forgetting, on FMNIST, DFA-same achieves 10% less average forgetting than BP in both split and permuted datasets in the Domain-IL scenario. This ranking is conserved when the experiments are performed with the MNIST datasets (see Appendix E) and CIFAR10 (see Table 2 and Appendix F). The advantage of DFA-same in the Domain-IL scenario is in accordance with the hypothesis that DFA-same learns close representations for different tasks. In fact, the Domain-IL has a unique output layer shared among all the tasks and this requires similar representations among different tasks in order to correctly classify the previous ones.

In the Task-IL scenario, the networks have the advantage of a dedicated output layer for every task. This is reflected in a better average forgetting for BP in the Domain-IL scenario by 8% and 22% for the Split and Permuted FMNIST datasets, respectively. On the other hand, the Task-IL scenario applied to DFA-same improves average forgetting only by 6% on the split dataset and becomes worse, with average forgetting from 45.8% to 50.1%, in the permuted-FMNIST. This trend (also observed in CIFAR10) causes DFA to perform worse than BP in the Task-IL split case, while being comparable to BP on the Task-IL permuted dataset[1]. The average forgetting trend of DFA-same can be explained in light of our hypothesis: if the output layer is specialized for every task, keeping the representations similar to each other can bring a disadvantage, as the previous representations are "overwritten".

Comparing Random Features and DFA in minimizing forgetting, we find that DFA can achieve a better Average Performance than a Random Feature model, both in the split and permuted datasets, while maintaining zero forgetting, just as RF, in the FMNIST. Specifically, DFA shows average performance at least 2% higher than RF (see table 1, column "Minimizing forgetting"). One example is shown in fig. 3, where DFA reaches 83% accuracy while Random Features reaches 80.9%. The fact that DFA reaches higher performances than RF means that the weights of the first two layers are updated, and the zero forgetting values of DFA means that this weight update can conserve the information learned during the previous tasks.

Overall, we find that DFA can embody different behaviours according to the value of its hyper-parameters. It can almost reach the same generalization performances as BP while displaying less forgetting in the majority of the cases. Furthermore, it can achieve zero forgetting like RF, while generalizing *better* than RF. Thus, DFA occupies a sweet spot between the plastic BP and the static RF.

### 3.2 Extension of DFA to the Task-IL scenario

We showed that the standard DFA cannot benefit from the introduction of a specialized output layer for every task (Task-IL scenario). We propose an extension of DFA where a different feedback matrix is used for every task: in fig. 4, we show that in this setting, the representations of one dataset are learned in different manifolds. According to our hypothesis, this phenomenon can be extended to the learning of different datasets allowing the network to learn new tasks with less interference from previous representations. We expect that the combination of a dedicated feedback matrix alongside a dedicated output layer can reduce forgetting in the Task-IL scenario.

We find that the main differences with respect to DFA-same are in the Average Forgetting measure and are in accordance to the hypothesis: when optimized for performance and in the Task-IL scenario, DFA-diff has $9.8\% \pm 2\%$ average forgetting in the FMNIST dataset and $2.9\% \pm 0.4$ average forgetting in CIFAR10. These

---

[1] The comparable value of average forgetting between DFA and BP in the Task-IL permuted case hinders the advantage of DFA-same for the forgetting of the earliest tasks. We show in fig. 3 the trend of forgetting in the first task, and we can notice a clear advantage. In appendix H we report the forgetting trends of all the tasks, where one can see that DFA-same forgets more tasks seen one or two epochs before, but is more stable for earlier ones.

values are around 20% lower than the one of DFA-same and 5% lower than BP's in the same conditions. When optimized for minimal forgetting, DFA-diff achieves up to 1.4 % ± 0.4 average forgetting in the CIFAR10 Task-IL dataset, which is the lowest value after RF.

In the Domain-IL scenario, on the other hand, DFA-diff has at least 3% more forgetting than BP and 6% more than DFA-same. This shows that DFA-same is more suitable in the Domain-IL scenario.

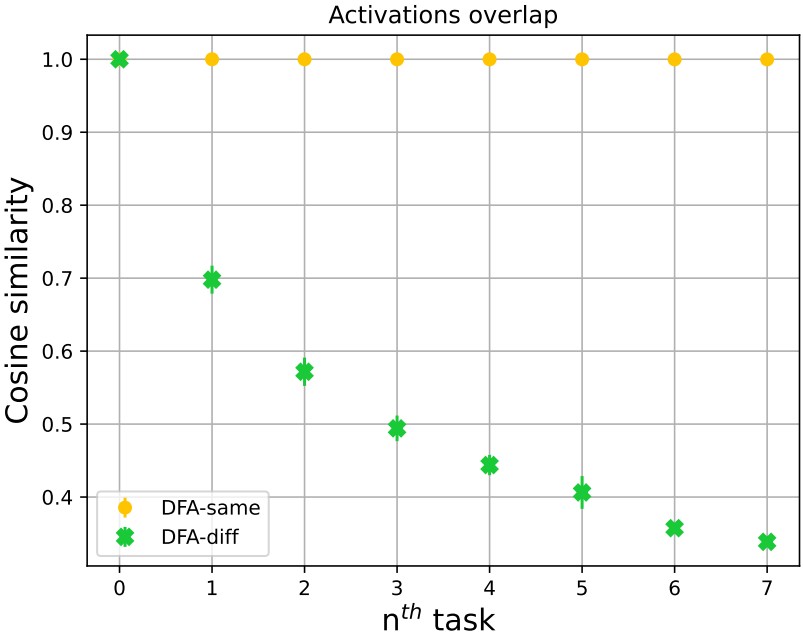

Figure 4: Cosine similarity of the change in activations due to the $n^{\text{th}}$ training vs the change since the first initialization. Values Averaged over the first two layers of the network and the images of the test dataset. The network is trained repeatedly on the same dataset (FMNIST) while the output layer is re-initialized at the beginning of every session. The overall change in activation in DFA-same is always aligned with the change taking place in every single training, while in DFA-diff it is less and less aligned with the changes due to subsequent trainings. The error bars refer to the sum of the standard deviation due to the averaging over different tasks for two different random seeds.

### 3.3 Comparing DFA with Elastic Weight Consolidation

In the FMNIST Domain-IL scenario there is an advantage of DFA-same over EWC of about 10% average forgetting. This could be due to the underlying hypothesis that in DFA-same the implicit regularisation given by the feedback matrix is kept constant. On the other hand, the direction of the regularisation factor in EWC is different for every new task, as explained in section 2. On the permuted dataset, EWC has a performance that is intermediate between DFA-same and DFA-diff. In fact, EWC can be considered a method in between DFA-same and DFA-diff because the "pulling forces" in the loss due to the feedback matrices will not be parallel to the ones of the previous tasks as in DFA-same, but not orthogonal as in DFA-diff, either. On the split datasets, on the other hand, EWC tends to be better than DFA-diff. In the Task-IL scenario EWC is the best method both when we evaluate on MNIST (see Appendix E) and FMNIST. In the Domain-IL it is the best only in the experiments on MNIST. Extensive hyperparameter tuning is required to improve the performance of EWC on FMNIST.

Additional results on split-CIFAR10 (see 2) confirm this trend. DFA-same has an advantage over EWC in the Domain-IL setting (8.1% Average forgetting wrt 16.0% ), while DFA-diff achieves lower Average forgetting (1.4% vs 2.5%) than EWC. In the second case, for DFA-diff, the result holds even in training conditions in which performance is prioritized: DFA-diff can reach the same Average Accuracy of 89.1% as EWC while

maintaining less forgetting (2.9% vs 17%). On the other hand, the advantage of DFA-same in the Domain-IL is always demonstrated in a condition of slightly lower Average Performance: even when both methods are tuned for maximum performance, DFA-same shows the advantage of 4.2% less Average Forgetting at the price of 0.6% less Average Performance. Moreover, the flexibility of DFA in modulating the plasticity-stability trade-off through the design of the feedback matrix variance may offer practical advantages over EWC, whose performance is more tightly linked to sensitive regularisation terms that must be tuned for each new task distribution.

## 3.4 Comparing DFA with BP with dropout

The empirical evidence presented in Table 2 indicates that dropout significantly improves the forgetting of BP in the Task-IL scenario, bringing average forgetting from $22.8 \pm 4.6$ to $7.3 \pm 0.4$ in Task-IL CIFAR10, while it is slightly worse in the Domain-IL scenario. This behaviour is also observed in DFA-diff, but there are important differences between DFA-diff and dropout. First, dropout uses masking in the *forward pass*, while the gradients are the true ones. In DFA, the alignment makes the feedback matrix to bias towards the aligned configuration of the weights in the forward pass, but there is no direct masking of any neuron. In the *backward pass*, there is a sort of gating or masking mechanism, but the process is even richer because it imposes either null or positive or negative updates (while the derivative of the activation function does not change the sign because it is always positive or zero). Secondly, dropout masks a different random subset of neurons for every minibatch, while the Feedback matrix is kept unchanged during the training.

## 3.5 Ablation experiments

In order to investigate the reasons behind the success of DFA in some of the CL scenarios, we look for the underlying mechanism that influences DFA's stability to forgetting the most. For achieving minimum average forgetting, we performed a grid-search over the hyper-parameters and we find that forgetting is reduced with smaller variance of the feedback matrix for both DFA-same and DFA-diff, see fig. 5. This impact can be easily explained by observing that the entries of the Feedback matrix are a multiplicative factor in the update rule of the layers 1 and 2 (see section 2, in the right column of equations 3 and 4); thus the smaller the values in the matrix, the smaller will be the update of the weights of these two layers. The output layer is updated exactly as in BP, so this layer is not affected by the variance of the Feedback matrix. In the limit of a matrix filled with zeroes, DFA approaches a Random Feature behaviour where only the readout layer is updated. We therefore performed an ablation experiment with BP where we reduced the learning rate of BP in the first two layers to mimic the effect of feedback matrices with reduced variance, while keeping the learning rate of the readout layer fixed and similar to the one used for DFA. We show the results of the ablated experiment in all the datasets in fig. 5. We found that lowering the learning rate of the intermediate layer indeed brings BP towards lower average forgetting and lower average performance, similarly to the effect of lowering the variance in DFA's feedback matrix. In fact, BP with reduced learning rates can surpass DFA's stability in some cases, for example in the Task-IL, permuted case. In all the other cases, BP-ablated can reach average forgetting similar to DFA, underlining the importance of variance of the feedback matrix for the stability of DFA. Crucially, DFA still retains higher accuracy for old tasks in the long run: As we show in Appendix H, Figure 13, the accuracy drop for the case of Task-IL p-FMNIST in DFA is bounded to a maximum of 59% while BP-ablated reaches 67% forgetting.

# 4 Conclusions

We have explored the potential of the biologically inspired learning algorithm Direct Feedback Alignment, to alleviate catastrophic forgetting in artificial neural networks. By comparing the performance of DFA and various baseline algorithms on several benchmarks, we found that DFA has the potential for continual learning. For example, we found that DFA-same can alleviate catastrophic forgetting better than BP and EWC in the Domain-IL scenario. This result is consistent with the first hypothesis, whereby DFA-same imposes weight alignment in the presence of different datasets. We saw that the networks are indeed able to exploit this for mitigating catastrophic forgetting. According to our hypothesis, DFA-same works as an

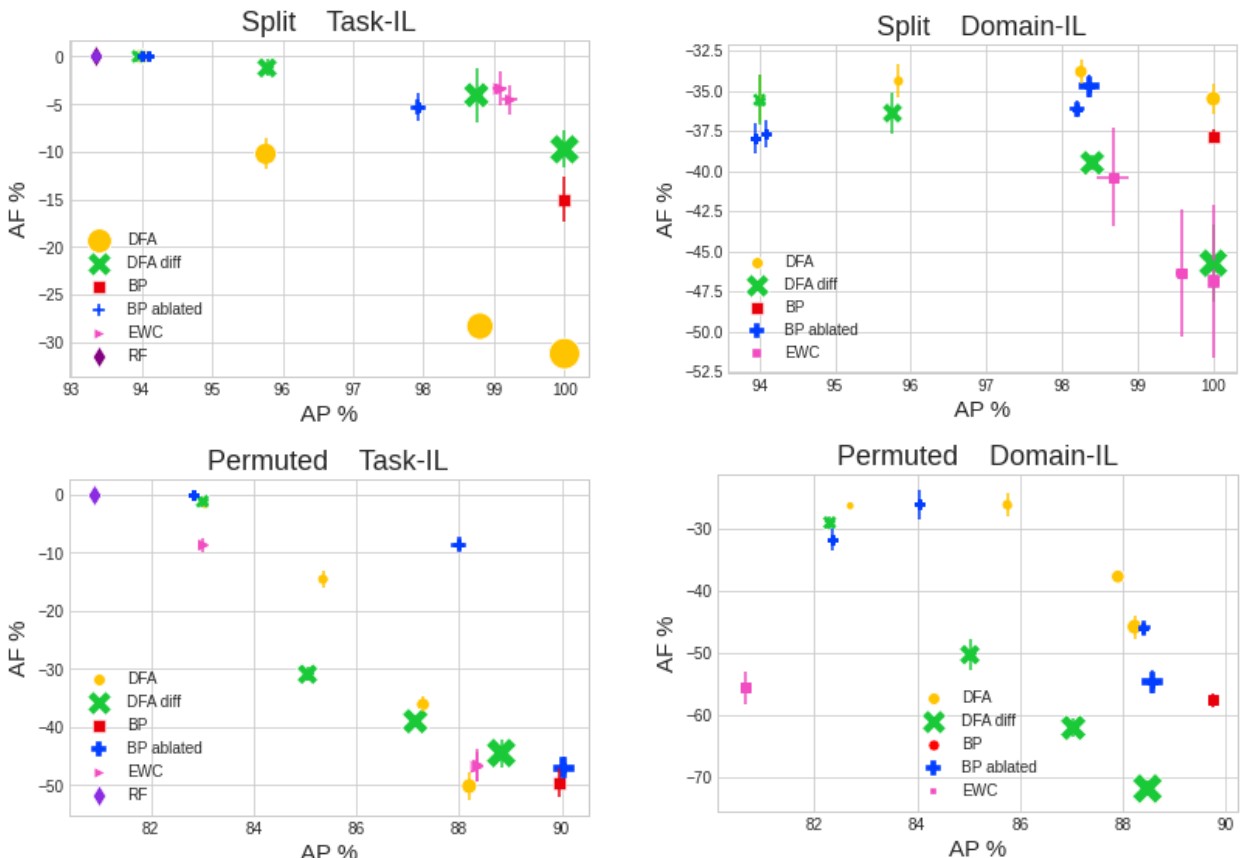

Figure 5: Average forgetting - average performance plots for all datasets (by rows) and scenarios (by columns). In each plot, the increasing size of the marker in DFA and DFA-diff indicates an increase in the variance of the feedback matrix (from $1.8 \times 10^{-3}$ to $1.8$) and for BP ablated an increase of the learning rate of the first two layers (from $10^{-8}$ to $10^{-3}$); the learning rate of the third layer is kept to $0.01$ to match the value used in DFA and no dropout is used. We can notice that for larger values of the feedback matrix variance, DFA can reach higher average performance at the price of smaller average forgetting (smaller dots moving towards the top-left part of the plots, which culminates in the RF regime). The same trend is followed by BP when the learning rate of the first two layers decreases (BP ablated).

implicitly regularised method like EWC, but with the advantage of keeping exactly the same constraints on the weights, while EWC adjusts the constraint of the weights after every new dataset encountered.

Empirical evaluation also shows that DFA with different feedback matrices for each task in the curriculum has an advantage with respect to EWC and BP in the Task-IL scenario. This is the architectural scenario in which the output layer is specific for every task. This allows learning the features on different manifolds in weight space, while still permitting an efficient encoding of the features for a successful classification of the different tasks. In continual learning, learning the features on different manifolds makes the learning of the new tasks not interfere with the separating hyperplane of the previous tasks, so it can be an advantage in the Task-IL scenario. The drastic improvements of DFA-diff in the Task-IL scenario with respect to DFA–diff in the Domain-IL scenario in contrast to the minor improvement of DFA-same in these scenarios corroborates our second hypothesis, whereby gradient alignment extends to the case of different datasets, allowing DFA-diff to learn in different manifolds. This approach can thus be employed for an effective mitigation of catastrophic forgetting. Finally, we tested gradient alignment in two specific experiments, and we find it visible in the case in which DFA is trained on the same series of datasets starting from different initialisations with the same feedback matrices ( Appendix A, panel C) and when DFA is trained on the

same dataset repeatedly but with different Feedback Matrices ( Figure 4).Taking inspiration from the impact of the scale of DFA's feedback matrix on forgetting, we designed BP ablated: backpropagation with reduced learning rates in the first two layers and we found that it has a significantly lower overall forgetting than DFA-same in the case of Task-IL permuted dataset. However, DFA-same retained its advantage with respect to BP and BP ablated in the Domain-IL scenario and for the oldest tasks in the Task-IL (Appendix H. The advantage of DFA-same over this method is further evidence that weight alignment is taking place and is beneficial against catastrophic forgetting beyond the impact of the Feedback matrix on the learning rate.

We noticed that in the split datasets forgetting is heavily dependent on the similarity of consecutive tasks (see larger error bars in fig. 5). For this, it would be beneficial to adopt curriculum learning. Moreover, a pre-processing of the dataset with pre-trained convolutional layers can replace the highly-variable distribution of pixels with more rationalized features. This strategy was already adopted in DFA (Crafton et al., 2019) and we extended the same idea to Continual Lerning: In our experiments on CIFAR10, we adopted the strategy of using DFA to train only the FC part of a pre-trained Convolutional network (Crafton et al., 2019) and we showed that the advantage of DFA-same and DFA-diff extend in this condition, rendering DFA competitive also in larger scale datasets and architectures. In the future, it will be interesting to investigate how the architecture of the networks, and specifically the width and the depth of the FC network influence the plasticity-stability trade off curves.

Our claims were supported by empirical evidence tested on the Class-IL scenario (Appendix I), where the task-ID must be inferred, and the second hypothesis was confirmed also by means of the Forward Transfer (Appendix D) measured on the Task-IL scenario.

The existing advantages of DFA can be extended, starting from the integration rules, as suggested in Della-ferrera et al. (2021). In this case, the firing of each neuron is allowed only if it receives at least $n$ positive inputs, while in the current implementation it is enough that one pre-synaptic neuron delivers a strong signal.

In conclusion, this work sets the foundation for employing DFA in Continuous Learning. We showed that it can both be a tool for regularization and can provide a sudden orthogonalization of the gradients without costly operations.

## 5   Acknowledgements

We thank Matteo Santoro for his valuable help in discussions and advice to structure the manuscript. We also thank Eszter Székely and Alessio Colucci for proofreading the manuscript.

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

## A    Degeneracy breaking of DFA

Previous work on DFA (Refinetti et al., 2021) and FA (Lillicrap et al., 2016) has shown that DFA brings the gradients and the weights to align with the feedback matrix. This phenomenon is called degeneracy breaking because among all the combinations of the weights that can fit the dataset, DFA selects one that is closest to the one aligned to the feedback matrix. **Gradient Alignment** and **Weight Alignment** were previously measured on the same dataset, starting from different initialization seeds. Here, we show the overlap of the weights (in panel A) and the overlap of the gradients (in panel B) among networks with the same/different initialization seed on the weight (yellow/purple in the first column of panel A) and the same/different feedback matrix (DFA-same/DFA-diff). In the training on Task 1 all the networks have the same feedback matrix.

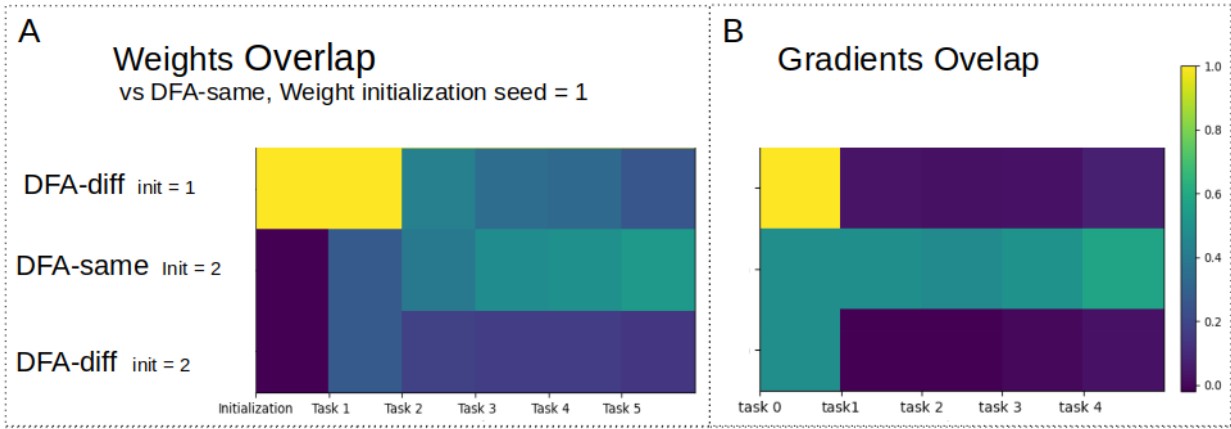

Figure 6:    **A**: Dot product of the weights at the end of training on the different tasks (averaged over the first two layers, Domain-IL architecture structure) of two networks trained with DFA-same or DFA-diff in contrast with a reference network trained with DFA-same. **Weight Alignment** can be appreciated mostly by looking at the second row, the case in which the networks are initialized from different seeds (seed 1 for the reference network and seed 2 for the network in the second row): the weights at first do not overlap, but thanks to the fact that the networks are trained with the same feedback matrix, the weights overlap more and more. **B**: Overlap of gradients averaged over two different checkpoints during the training on the same dataset (splitFMNIST, Task-IL architecture structure) of two networks trained with DFA-same or DFA-diff in contrast with a reference network trained with DFA-same. **Gradient Alignment** is more visible in the first and the third row: the gradients in panel B become orthogonal (overlap equal to 0) starting from the second task, as soon as the feedback matrix changes, irrespective of the fact that the networks are initialized in the same way (first row) or if the networks are initialized differently (third row).

## B    Pre-training on Imagenet is allowed?

Convolutional networks layers are complex architectures that are specialized for image classification Celeghin et al. (2023). As we discussed in the introduction, DFA fails at training convolutional layers, so in order to scale to extensive dataset we will adopt 3 layers of DFA after convolutional layers pre–trained with BP Han & Yoo (2019). In Continual Learning, one of the requirements is not to use information of the training tasks to pre-process the dataset (like in the selection of the hyperparameters). So we include an analysis in this section if pre-trained convolutional layers on imagenet carry any feature about CIFAR10.

The convolutional architecture used is a VGG11 architecture which is a relatively small convolutional network composed of 8 convolutional layers ( 3x3 filters with stride 1 and padding 1) followed by Batch Normalization, ReLU activations and Max-pooling ( 2x2 filters and stride 2). The convolutional part is followed by two FC layers (4096 hidden size, dropout 0.5) and one Softmax layer (for a total of 3-layer FC network). The FC network used when DFA is used has a hidden size of 1096 and 4 layers without dropout.

This experiment is characterized by the fact that the softmax layer at **the end of the FC network is not trained**. This doesn't allow DFA to align its last–layer weights, and thus the Gradient Alignemt don't take place, leading to DFA to be stuck in a training trajectory that focuses on the Weight Alignment of the updated layers, without any connection with learning (dashed and dotted lines in Figure 7).

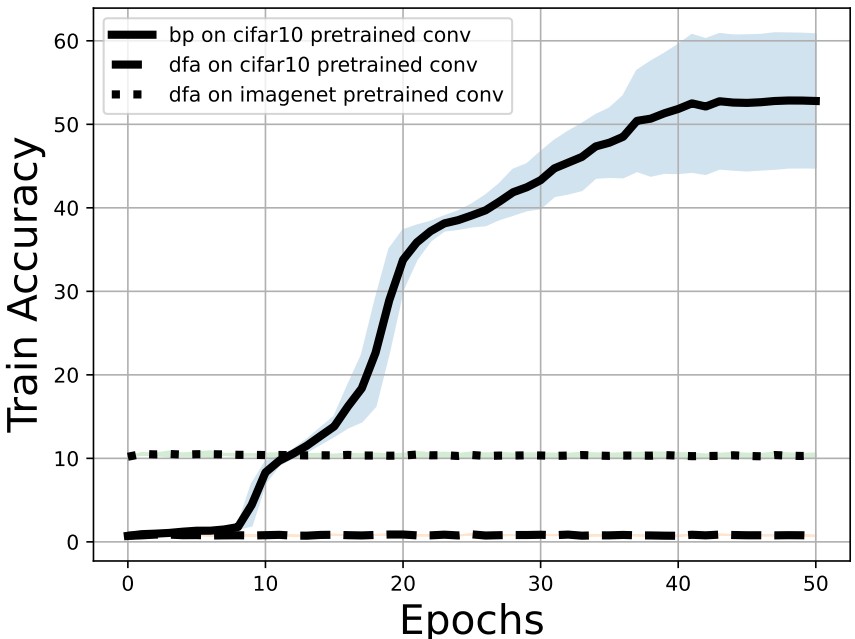

Figure 7: Experiment of the pre–trained fixed convolutional layer followed by 3 FC layers and one fixed softmax layer. The shadow is one standard deviation of models trained with hidden size of 4096 and different lr, momentum, batch size computed in grid combinations among the values of lr=[0.1,0.01,0.001]; wd = [0, 1e-5], mom = [0,0.5]; batch size =[30,512].

While BP (solid line) updates the inner layers of the FC network in a meaningful way. The result of the this experiment is making visible that the **pre–training on ImageNet** offers DFA with features for which learning starts at 10%, which is the Random Guess Accuracy. When pre-training on a different dataset, the model has learned more general feature representations that aren't specific to the target dataset. Therefore, when transferred, the model starts with some general knowledge, leading to random but non-zero accuracy at the start. Erhan et al. (2010).

Instead, pre–training on CIFAR10 (the same dataset using in the experiment), the Test Accuracy of DFA is stuck at 0%. This can be explained by the opposite argument: the convolutional layers have highly become specialized for that dataset. During fine–tuning, the weights are already adapted to the target data distribution, so the model's initial predictions might be incorrect for the new task (here, there are three layers of random projections mixing the features), causing the initial accuracy to drop to zero. This experiment is useful to prove that using pre-trained convolutional layers on IMAGENET does not carry dataset-specific information about CIFAR10, thus they can be used for the Continual Learning evaluation of DFA on CICAR10.

## C   Similarity Measures of datasets

In fig. 1, panel B, we analyze the similarity of the datasets arising from the division into the different tasks. We take in the rows all the images of one class in the dataset corresponding to task1 (and their pixels in the columns) and we compare it to the matrix of images of the same class in the next dataset. We average over all the couples of datasets and over the different classes. The similarity measures we apply (described below) measure the similarity from a merely geometrical perspective and we find that in this point of the tasks have

classes more similar to each other in the permuted dataset. By construction, the reshuffle of the pixels does not change the mean and the standard deviation of the distributions of the pixels. Nevertheless, the split datasets is easier because it has only two classes and in the Task-IL learning the difference in distribution can be an advantage. In fact, the average forgetting in the Task-IL scenario are overall much smaller than the average forgetting on the permuted dataset. On the other hand, the permuted dataset has only slightly more forgetting in the Domain-IL scenario in the case in which the models are optimized for performance.

we use the following similarity measures:

1. Cosine similarity: Dot product of the matrices reshaped into vectors and subsequently normalized by the L2 norm of the two vectors.

2. Canonical Correlation Analysis (CCA) (Hardoon et al., 2005): Similarity measure which is invariant to affine transformations (any linear invertible transformation including scaling, rotations, translations).

3. Centered Kernel Alignment (CKA) (Kornblith et al., 2019): Similarity measure which is invariant to orthogonal transformations, which is a subset of the invertible linear transformations.

Cosine similarity measures pairwise angles of the images in the same class among the different datasets and it is sensitive on the local alignment. This can become higher if, for example, the images are the same but are misaligned. Instead, it can be low if the images are aligned. This measure gives a measure of how much the first layer of the network has to be updated among the different tasks because the less the row-inputs are aligned, more the first layer will cause catastrophic forgetting. Trough this measure, we display something that is not intuitive: the permuted protocol in the FASHIONMNIST dataset has classes that are less aligned with respect to the split protocol even though the pixels have been reshuffled.

CCA is 1 when the two matrices are linearly invariant (they are linked by any affine transformation) and 0 when they are not. In other words, CCA is filtering out the best linear transformation that maximize correlation between the two datasets. This is a measure of how much the second layer in the network will be faced with differences in the tasks, assuming that the first layer is able to learn the best linear mapping ( keep into consideration that the first layer is the one that starts learning first in DFA (Refinetti et al., 2021)).

CKA is 1 in the case that the two matrices are orthogonally invariant (can be linked by a rotation). This invariance can be seen as a filter to the best non-linear mapping of the inputs.

With the plot in fig. 1 we show that the permuted dataset is composed of images with distributional similarity among the tasks.

## D  Forward transfer

In addition to the Average Performance and Average Forgetting, another evaluation metric as the Average Forward Transfer have been introduced by Chen et al. (2023). This is the average accuracy on every unseen task after training a linear classifier on the representations learned at the previous stages and is sought for the complete evaluation of the Task-IL scenario.

We performed this measurement on DFA–same and DFA–diff trained on split FashionMNIST.

As we can see in Figure 8, the forward transfer can reach as much as 75% when the fine-tuning is performed with 500 test data points. Without any fine-tuning, the Accuracy would be lower than 50% which is the random guess baseline. DFA–same displays a slight advantage over DFA–diff, especially in the case of fine–tuning over 100 data points.

Observing the forward transfer of the un-trained heads (see Figure 9, DFA-same on the left and DFA–diff on the right), we notice an interesting pattern: In DFA–diff, the accuracy on unseen tasks depends solely on the type of representations used, while in DFA–same the different tasks under test vary in Forward transfer. More precisely, the Accuracy varies with a monotonically decreasing gradient from up to down along the

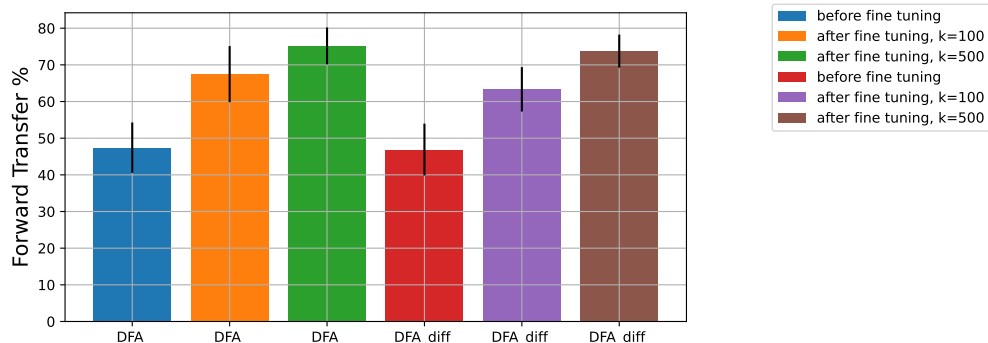

Figure 8: Mean Accuracy on Split-FashionMNIST on unseen tasks after training a classifier on top of the representations learned at any stage. The error bars are the standard deviation over 3 random seeds and the various tasks. Hyperparameters: 1000 epochs, lr=0.01, wd=0, mome=0, var(F)=0.1.

columns (from oldest task to the most recent), and correlate with the pattern of DFA–diff along the rows. This could be explained with the fact that Forward Trasfer between adjacent tasks are prone to have the same value. This is visible by observing the upper diagonal in the DFA–same (left panel, Figure 9); while the forward transfer between distant tasks does not reflect this, and instead with DFA-diff the features useful for the future tasks are not affected by training on the previous tasks. This is in accordance to our second hypothesis.

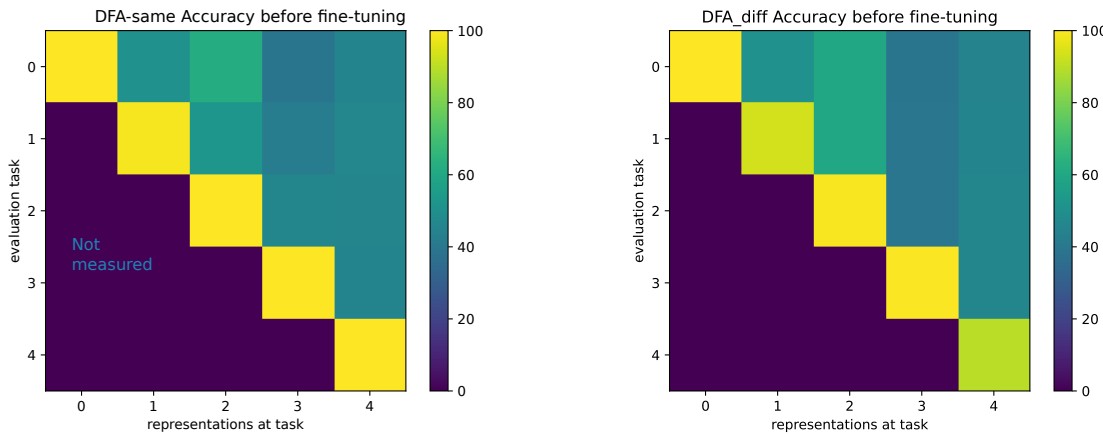

Figure 9: Test Accuracy on Task-IL split-FashionMNIST of the unseen tasks before fine-tuning the head. Same hyper-parameters as above.

# E    Results evaluated on MNIST

The plots displaying the results of the methods applied to the MNIST datasets in the Task-IL scenario can be found in fig. 10. The points in the circle indicates the Split-MNIST, otherwise they refer to the results on the Permuted-MNIST.

# F    Convergence of DFA: impact on Continual Learning

In this section we show the performances of a 3-L FC network trained with DFA for 50 epochs vs 1500 epochs. We find that, as expected, the one early in the training behaves closer to a Random Feature model,

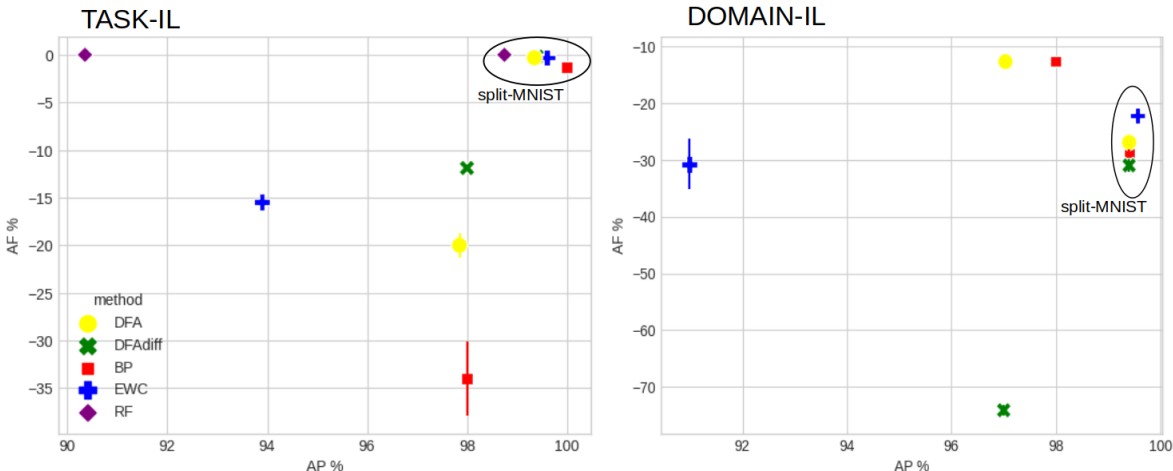

Figure 10: We can notice that the results are consistent with the results on F-MNIST: In the Task-IL split-MNIST DFA lays in an intermediate position between RF and BP, slightly behind EWC; In the permuted Task-IL, DFA-diff has less forgetting than all the other methods while achieving the same AP of BP. In Domain-IL (panel on the right), DFA-same is comparable or equal in AF to BP while DFA-diff has the worst performances; EWC is in an intermediate level between DFA-same and DFA-diff on the permuted dataset while is superior to DFA on the split datasets.

being subject to the "alignment" phase described by Refinetti et al. (2021). In the Task-IL scenario, DFA–diff is in the same radius as BP from the corner of desirable performances: the top-right, with highest AP and lower AF. In the Domain-IL scenario, DFA-same is in a lower radius with respect to BP from the upper-right corner. Future work is needed to understand how the minima found at different levels of over- or under-fitting are correlated within one task and between the different tasks.

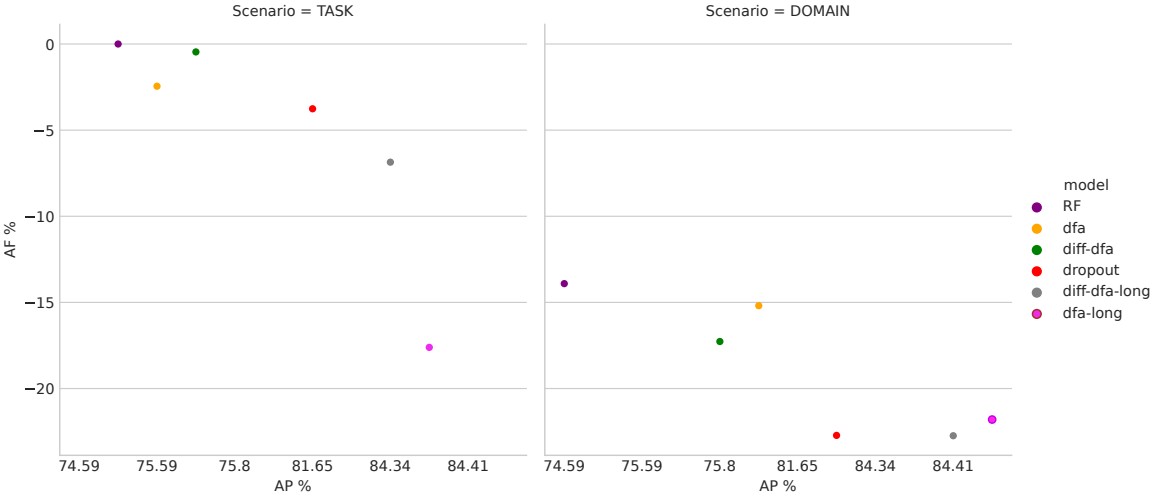

Figure 11: AP and AF measures on 3-Layer FC network on the CIFAR10 dataset. Hyperparameters: Random seed = 1, momentum=0.3, wd=0.0001, lr=0.01, hidden size= 1000, batch size=512.

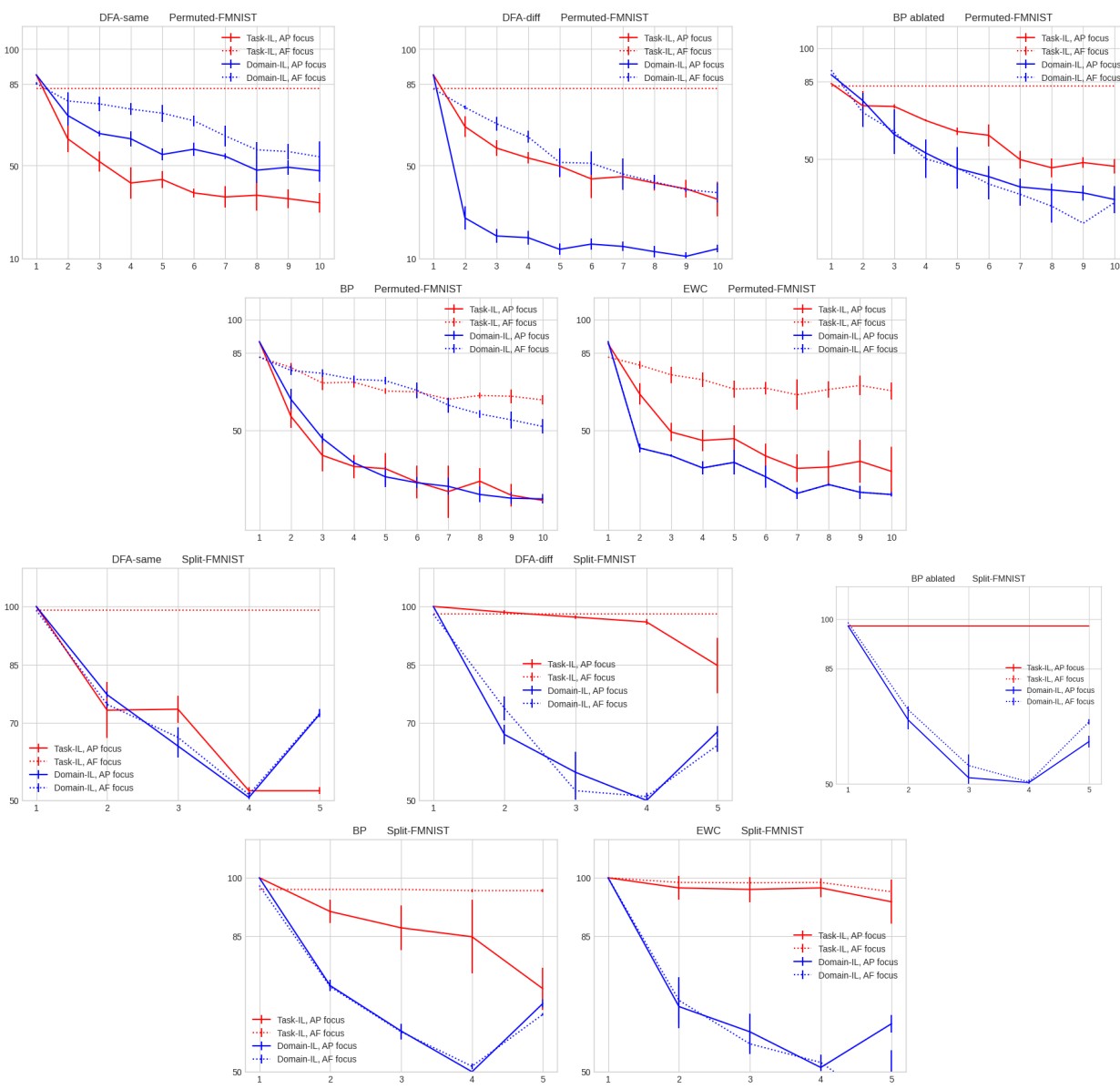

Figure 12: Complete illustration of forgetting the first learned task for all methods (Accuracy % on the y-axis and task number on the x-axis). The different figures display one method at a time in the following order: DFA-same, DFA-diff, BP-ablated, BP, EWC; permuted FMNIST above and split FMNIST below. The dotted lines are the ones optimized for minimizing forgetting. In EWC permuted FMNIST, the blue dotted line coincides with the blue solid line.

## G  Random Seed impact on the accuracy of the firstly learned task

## H  DFA-same versus BP and BP-ablated at increasing tasks

In order to answer the question, "Is DFA mitigation for catastrophic forgetting solely due to the small learning rate in the first two layers?" we propose a deeper comparison with BP ablated in the context of the permuted dataset. The same results apply to the split dataset. In fig. 13 below, we display the accuracy of all the tasks during the CL pipeline (training one task at a time and always monitoring the accuracy of the other tasks; in the case of Task-IL scenario, either for training and for testing one task, the corresponding

output layer is used). We compare the case in which BP ablated has the most similar AF to DFA-same and we find that DFA has the tendency of forgetting more than BP and BP ablated after few tasks but is more stable in the long run, for example after 6 tasks in the Task-IL and after 1 or 9 tasks in the Domain-IL scenario; the black and the grey lines in the plot of DFA-same visually show the point where the advantage starts. We conclude that a small learning rate in the first two layers is not always enough to reproduce DFA-same stability and this advantage in the long run might be due to the alignment of the weights.

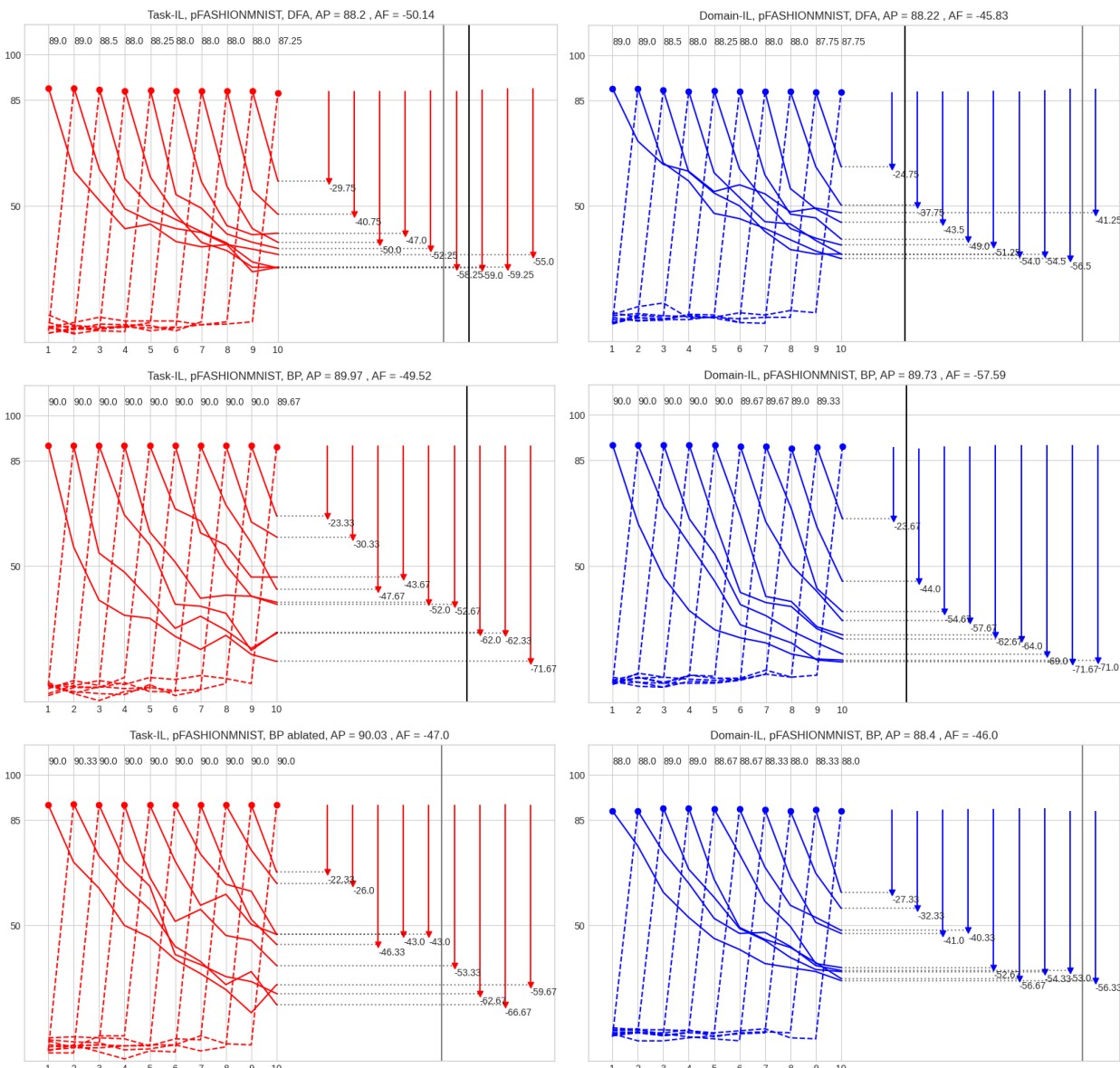

Figure 13: Accuracy interpolation lines along the tasks. On the left (red lines) we show the results for the Task-IL scenario, we can notice that even though DFA-same has higher average forgetting than BP and BP-ablated, it has lower forgetting for the first three (vs BP) or four tasks (vs BP ablated): DFA-same retains the accuracy better than BP and BP-ablated in the long run, while it forgets more in the shorter run. This is also true in the Domain-IL scenario (blue lines, on the right) in which DFA has better average forgetting than BP and worse compared to BP-ablated.

# I Class-Incremental Learning

Catastrophic Forgetting is an issue in real-world cases mostly in the challenges reflected by the Class-IL scenario, where the network must infer the task ID alongside the correct class. This can be reformulated as a broader classification task in which the number of classes grows as new tasks are added. We evaluated DFA in the class-IL scenario without any hyperparameter optimization specific for this case and without the strategies that could help to improve in this scenario (Curriculum learning, class replay, ecc). In Figure 14, we report the Test Accuracy of the network computed by aggregating the outputs of the 5 heads trained as in the Task-IL protocol. The aggregation of the outputs was performed by simply stacking the outputs (solid line) or adding a softmax layer to re-normalize the outputs. With the first procedure, there is a peculiar behaviour of DFA-diff: the second head has increasing accuracy along the stages. This means that the training on the second dataset rendered the network very sure about the classification, and the following tasks did not disrupt this signal. This can be achieved by DFA-diff by learning the following tasks changing the representations along orthogonal manifolds. The solid lines in DFA-same that exhibit a strong classification certainty, instead, decrease after the network is trained on the subsequent tasks. This can be explained by the fact that DFA-same changes the representations in a disruptive way, and this was the reason why DFA-diff was designed for the Task-IL scenario. By re-normalizing the outputs with an additional softmax function, we can notice that DFA-same achieved performances above the random threshold of 10% only for two tasks, and only one of them is classified above this threshold at the end of all the tasks. At this point, on the other hand, DFA-diff has non-trivial performances on two tasks.

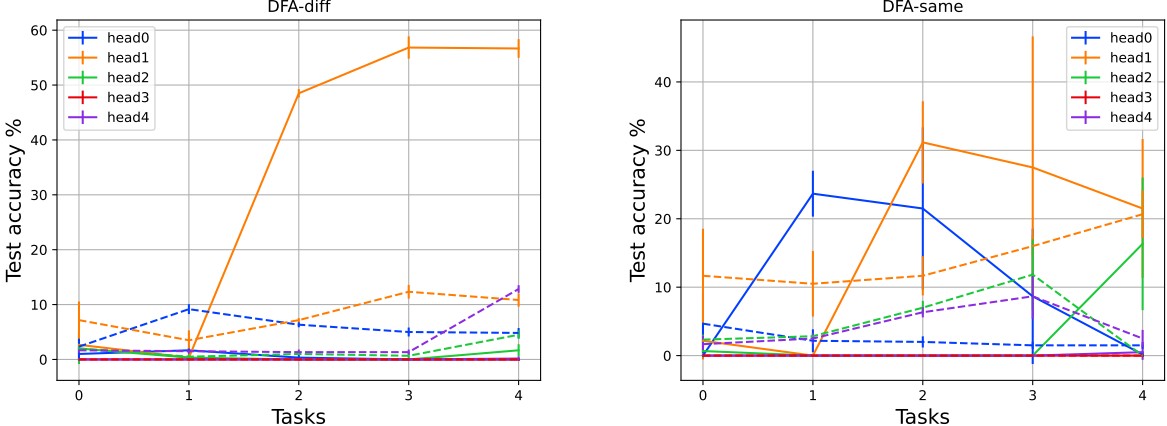

Figure 14: Evolution of the Test Accuracy evaluated by aggregating the 5 heads trained ( only after the corresponding task on the x axis) as in the Task-IL scenario. Dashed lines: the aggregation was performed before the softmax layer (so one additional Softmax layer re-normalizes all the outputs of the different heads). Solid line: the aggregation of the output was performed after the softmax layer, so the prediction is performed by selecting the head with the strongest activation.

