# OpenReview forum: "Exploring the potential of Direct Feedback Alignment for Continual Learning"
_TMLR — Accepted by TMLR_

### Review · Reviewer_N6hZ · 2024-08-07

**Summary Of Contributions:**

The authors explore the effects of using direct feedback alignment (DFA, i.e. using random update matrices directly from the output to intermediate layers) in continual learning. On MNIST, Fashion-MNIST in domain and task-incremental settings, they find that sometimes DFA outperforms vanilla backpropagation (BP), random features (RF), and elastic weight consolidation (EWC), both in terms of average performance and average forgetting. They also try two versions of DFA: DFA-same, where the feedback matrices are the same across tasks, or DFA-diff, where they are randomly sampled per task. They show that each one has its trade-offs: while DFA-same is good for domain-IL, DFA-diff is better at task-IL. Finally, they show that lowering the learning rate of hidden  layers in backprop can be seen as a proxy between DFA-diff and DFA-same, achieving a balance between the two.

**Audience:**

Yes

**Claims And Evidence:**

Yes

**Requested Changes:**

# Requested changes
Given the weak experimental setup and the lack of theory, I am not sure whether this submission is solid enough to serve as a foundation for future research. I would recommend the authors consider each of the following points of improvement:

* Providing evidence that results hold in modern architectures and benchmarks
* Theoretically confirm their claims
* I would also deepen on the effects of DFA convergence for continual learning.

**Strengths And Weaknesses:**

## Strengths and weaknesses
**Strenghts**
* I found the idea: "because DFA always converges to the same minima, it should be more robust to task interferences", very interesting and worth exploring.
* The realization that DFA is similar to BP with lower learning rate on intermediate layers is also interesting.
* Overall the text is well-written and easy to understand.

**Weaknesses**
* Given how toyish the experimental setup is, it is not clear whether results would generalize. Note [B] discourages the use of permuted MNIST, and since [C], the community has shifted towards more holistic evaluations rather than focusing on performance and forgetting. This could be compensated with a theoretical analysis showing, e.g. that lowering the learning rate on intermediate layers is equivalent to DFA, however such analysis is missing. I think this is the major weakness of this submission.
* The authors add dropout to improve their baseline, however, dropout has been shown to have some effect on continual learning, so it might interfere with the authors' findings [D].
* DFA for continual learning has already been explored in the past (implicitly), particularly as a meta-learning problem [A], which is not mentioned in the submission.
* Actually, it would be interesting to explore the relationship between dropout and DFA-diff.

**Minor**
* I encourage the authors to use a vectorial format to embed their figures in $\LaTeX$ so that they do not appear pixelated.
* The authors use the word "task" and "stage". I would just use one word and define it clearly early in the text.
* Footnote 2 contains ??
* Page 11 contains ??


[A] Lindsey, Jack, and Ashok Litwin-Kumar. "Learning to learn with feedback and local plasticity." Advances in Neural Information Processing Systems 33 (2020): 21213-21223.

[B] Farquhar, Sebastian, and Yarin Gal. "Towards robust evaluations of continual learning." arXiv preprint arXiv:1805.09733 (2018).

[C] Veniat, Tom, Ludovic Denoyer, and MarcAurelio Ranzato. "Efficient Continual Learning with Modular Networks and Task-Driven Priors." International Conference on Learning Representations.

[D] Mirzadeh, Seyed Iman, Mehrdad Farajtabar, and Hassan Ghasemzadeh. "Dropout as an implicit gating mechanism for continual learning." Proceedings of the IEEE/CVF conference on computer vision and pattern recognition workshops. 2020.

---

> ### Author Response · Authors · 2024-12-02
>
> ## Weaknesses
> > This could be compensated with a theoretical analysis showing, e.g. that lowering the learning rate on intermediate layers is equivalent to DFA
>
> Thank you for this suggestion, it is a crucial point to be discussed.
> Our claim is that DFA with a small feedback matrix is not equivalent to lowering the learning rate on intermediate layers. Although there is an algorithmic similarity between the two methods, we show that DFA brings additional benefits. A theoretical analysis of this claim appears very hard: analysing the dynamics of deep networks beyond two layers, which is the only setting in which this point could be studied, remains an open problem even in the simplest case of supervised learning, so such an analysis is beyond the scope of the present work.
>
> > Note [B] discourages the use of permuted MNIST
>
> We agree that the permuted is a task that requires learning without relying on the previous task's knowledge, and this is the only case in which BP ablated has higher AF with respect than DFA (Permuted, Task-IL scenario and Domain-IL scenario as we expand in Appendix H). This kind of analysis is useful if one seeks at combining DFA with convolutional pre-trained layers, in which useful features are already extracted (Crafton et al FNINS 2019).
>
> > DFA for continual learning has already been explored in the past (implicitly), particularly as a meta-learning problem [A], which is not mentioned in the submission.
>
> Thank you for your comment, we found some interesting commonalities between our work and the content of this reference: they also limit the plasticity of the network to the last layers. We added this reference in the introduction.
>
> > The authors add dropout to improve their baseline, however, dropout has been shown to have some effect on continual learning, so it might interfere with the authors' findings [D].
>
> Dropout was used only to have a stronger baseline and was not used in DFA nor in BP-ablated. In the experiments on FMNIST, we adopted dropout in BP when it was beneficial, and we used dropout in a mild form: 0.2% in the first layer and 0.5% in the other intermediate layers.
> To address this exhaustively, we added a section on it and explicit versions of BP and BP dropout in the CIFAR10 experiment.
>
> >Actually, it would be interesting to explore the relationship between dropout and DFA-diff.
>
> This is indeed an interesting point to expand, thank you for bringing this up. We expanded this point in section 3.3.
>
> ## Minor
>
> - We changed all the figures, converting them in a vectorial form.
> - We replaced "stages" with "tasks".
> - We corrected the two broken references, that were leading to ??. Thank you for pointing them out.
>
>  # Requested changes
>
> >Providing evidence that results hold in modern architectures and benchmarks
>
> We corroborated our claims with experiments on CIFAR10 and using the features extracted by modern architectures such as VGG11 (See Table 2): DFA-same is the best method in the Domain-IL and DFA-diff is the best method in Task-IL. This extends our results to a more challenging benchmark (CIFAR10 with three colours) and more articulated architectures.
>
> > Theoretical confirmation of our claims
>
> Thank you for raising this concern, it gives us a chance to clarify the theoretical _roots_ of our approach.
> Our approach to continual learning is firmly rooted in the theoretical analysis of DFA performed by Refinetti et al. (ICML 2021), who gave a mathematically precise description of the Weight Alignment (WA) and Gradient Alignment (GA) between networks and their feedback matrices, and whcih crucially identified the degeneracy breaking mechanism which is the basis to our approach.
> Our work provides empirical evidence that degeneracy breaking has important ramifications for Continual Larning. A more comprehensive analysis of DFA in continual learning could be carried out by the combination of the existing teacher-student analysis of DFA (Refinetti et al. ICML 2022) and the teacher-student analysis on Continual Learning (Lee et al. ICML 2021, Lee et al. PMLR 2022).
>
> > Deepen on the effects of DFA convergence for continual learning
>
> DFA has different stages of convergence as shown by  Refinetti et al. (ICML 2022): when the network is in the "alignment" phase, it behaves more rigidly, closer to a Random Feature model. When it is in the "memorization" phase, it diverges from the alignment towards the solution that fits the dataset. We updated the paper to show this phenomenon, see Appendix F.

---

### Review · Reviewer_rLU6 · 2024-08-16

**Summary Of Contributions:**

The authors study the impact of training with Direct Feedback Alignment (DFA) on catastrophic forgetting.

They compare DFA with backpropagation (BP) and other methods on various datasets derived from MNIST and FMNIST, and on both a task--Incremental Learning (task ID is known, different output head for each task) and a domain-Incremental Learning (no task ID, same outputs for all tasks) conditions.

They also introduce a variant of DFA, called DFA-diff, in which different feedback matrices are used for each task (which of course requires knowing the task ID).

Results are mixed. Standard DFA (DFA-same) has somehwat less forgetting than BP in Domain-IL. However, DFA-same actually has more or same forgetting than BP on the task-IL, when task ID is provided.

DFA-diff reverses this pattern. It has less forgetting than BP for Task-IL, but more forgetting than BP for Domain-IL (though IIUC DFA-diff requires knowing the task ID).

These results vary depending on whether one optimizes for performance, or minimize forgetting; the exact difference between these two conditions is unclear (see below).

The authors also compare DFA with Elastic Weight Consolidation, a standard continual learning method. Again, results depend on settings and conditions.

**Audience:**

Yes

**Broader Impact Concerns:**

I do not see any broader impact concerns.

**Claims And Evidence:**

Yes

**Requested Changes:**

Most important:


- The BP and DA equations on page 3 need to be rewritten more carefully (e.g. it is not true that dW2 = dJ/da2! Eq 3, right  should have dJ /da2 instead of dJ/da1, etc.)

- More importantly, Equation 4, right suggests that DFA uses the top-level error "e" for all layers of the network? This seems contrary to Nokland's original paper, equation 7, which explicitly uses B1 to backprop da2 rather than e  ? Can the authors confirm what exactly their method does?

- Figures 3 and 4 need some error bars, and the number of seeds used should be specified!

- What exactly is the difference between "optimized for performance" condition and "minimizing forgetting" condition? What parameters are varied? Are the same parameters used for all methods in each condition?

Others:

- What exactly does "degeneracy breaking" mean in the introduction?

- The description of permuted FMNIST and split FMNIST seem swapped. It is the former that uses dramatically different (randomized) representations of the same input. Also, the inputs do not use different modalities- they;re all images of the same size?... Something is off there.

- Footnote 2 has a missing reference.


- The authors note that for task-IL, DFA-same can have better performance than "random features" (i.e. frozen network except for the output layer, IIUC), while preserving 0 forgetting. However, IIUC from Table 1, under this training regime of "minimizing forgetting", DFA-same can have more forgetting than BP in the domain-IL setting! (This should be explicitly mentioned in the text, unless I missed it)

- In the Conclusion, we read: "The drastic improvements of DFA-diff in the Task-IL scenario in contrast to the minor improvement of DFA-same in this setting"... but DFA-diff has only minor improvement over BP, and DFA-same can actually be worse, according to table 1 ! Please change the language here.

**Strengths And Weaknesses:**

- The problem of catastrophic forgetting is important. To my knowledge, studying the effects of DFA specifically for catastrophic forgetting is novel. The experiments seem well made, though see below.

Weaknesses:

- The description of the methods are insufficient or just plain wrong, so it is not clear exactly what was measured. See below.

- The novel DFA-diff method requires one new set of backward weights for each task. But then why not also have a different set of feedforward, trainable weights for each task, which would completely eliminate catastrophic forgetting? This would only require 2x the storage over what DFA-diff already requires, which does not seem a big problem.

---

> ### Author Response · Authors · 2024-12-02
>
> ## Weaknesses
> > The novel DFA-diff method requires one new set of backward weights for each task.
>
> Thank you for raising this point, we edited the methods section to clarify this. In DFA-diff, there is not explicitly a new set of backward weights. What DFA-diff requires is a new feedback matrix for each task that is used in the backward pass to map the error to the activations of each layer. Since we use the same hidden size for every layer, the dimension of this matrix is output size times the hidden size times the number of layers.
>
> ## Requested changes
> >The BP and DA equations on page 3 need to be rewritten more carefully
>
> Thank you for spotting this typo out, we corrected equation 3.
> >This seems contrary to Nokland's original paper, equation 7, which explicitly uses B1 to backprop da2 rather than e ?
>
> Here is the clarification: Nokland's equation 7 refers to Feedback Alignment (FA), which is not to be confused with Direct Feedback Alignment (DFA), described instead in equation 8 of Nokland's original paper, which is the algorithm we consider here.
> >Error bars in Fig 3 and 4
>
> We updated Figure 3 and 4 to include the errorbars.
>
> > What exactly is the difference between "optimized for performance" condition and "minimizing forgetting" condition?
>
> Thanks for pointing this out, we now realise that the term 'performance' is perhaps too generic to describe what we mean. We changed the column description; we meant 'maximising AP' and in this case the hyper-parameters are chosen in such a way that the networks fit as much as possible the training set at hand. In the column described by 'minimizing forgetting', instead the hyper-parameters are chosen to minimize AF.
> > What parameters are varied?
> > Are the same parameters used for all methods in each condition?
>
> Thank you for the question, we realized that we only described the parameters of
> the model selection for DFA in the caption of Figure 4, which is after the
> table. We modified the methods section to include this information.
> The hyper-parameters varied are described in the methods section, in the paragraph "Baselines and hyper-parameters tuning".
> The architecture (number of layers and hidden size) and the batch size are the same among all the methods compared. What change is dropout (used only for BP), learn rate (EWC, BP), feedback matrix variance (DFA).
> In DFA the learning rate is kept fixed to 0.01 while the variance of the feedback matrix is selected among the range of 1 and 1e-8.
> ## Others
> > What exactly does "degeneracy breaking" mean in the introduction?
>
> The loss has multiple global minima and the solution is not unique. This is due to the fact that multiple parameter configurations yield the same output.
> With BP, due to the randomness of the initialization of the weights and the randomness introduced by the minibatches of the the data, the solution reached is a different one every time you run the algorithm.
> DFA breaks this multiplicity of solutions thanks to the alignment with the feedback matrix. \cite{Refinetti2021} described that DFA operates trough two phases: Alignment, then memorization phase.
> > The description of permuted FMNIST and split FMNIST seem swapped. It is the former that uses dramatically different (randomized) representations of the same input.
>
> We meant to emphasise that the average and standard deviation of all pixels are not affected by permuting, but we now realise this is confusing. We clarified this in the main text
> > Also, the inputs do not use different modalities- they’re all images of the same size?
>
> Very right, modality is not the correct word for what we meant, thank you for spotting it and giving us a chance to correct it. The subsets in which the whole dataset is divided into are binary classification tasks which input have different \textbf{statistical characteristics}, for example the mean, standard deviation and mode. We used improperly the word modality instead of "with different modes". By mode, in this sense, we mean the most frequent pixel distribution.
> >Footnote 2 has a missing reference.
>
> Thank you, we corrected this.
> > For task-IL, DFA-same can have better performance than "random features" while preserving 0 forgetting. However, from Table 1, under this training  regime of "minimizing forgetting", DFA-same can have more forgetting than BP in the domain-IL.
>
> Thank you for asking this clarification, in the Domain-IL DFA can reach 0 forgetting but at a higher price of AP. We reported in the table the minimum forgetting given the AP is larger than Random Features. This is clarified in the caption of the table.
>
> > .. but DFA-diff has only minor improvement over BP, and DFA-same can
>  actually be worse, according to table 1
>
> Good point, we updated the description of the phenomenon we observe.
> In general, we meant improvement with respect to the methods themselves in the two scenarios and not with respect to BP. We clarified the text.

---

> > ### Comment · Reviewer_rLU6 · 2024-12-02
> >
> > > In DFA-diff, there is not explicitly a new set of backward weights. What DFA-diff requires is a new feedback matrix for each task that is used in the backward pass to map the error to the activations of each layer. Since we use the same hidden size for every layer, the dimension of this matrix is output size times the hidden size times the number of layers.
> >
> > 1- The difference between "new set of weights" and "new feedback matrix with exact same dimensions as weights" is not obvious to me.
> >
> > 2- The main question is not addressed. If DFA-diff requires having access to task IDs and storing a new set of feedback matrices (one per layer) for every new task, then one might simply store a different set of feedforward weights for each task instead, for identical cost (IIUC), entirely removing the problems associated with continual learning. Thus it is not obvious to me what benefit is provided by DFA-diff.
> >
> > If my understanding is wrong I would welcome the authors' correction.

---

> ### Author Response · Authors · 2024-12-03
>
> Thank you for engaging with our replies.
>
> We are happy to have the chance to clarify the technical advantage of DFA. First of all, we emphasise that the DFA feedback matrices are _much smaller_ than the actual weight matrices, since one of their dimension is always the output layer.
>
> Pheraps the misunderstanding was rooted here:  The new feedback matrix has the dimension proportional to the __activations__, which is not equivalent to weights.
>
> To make it more clear:
> In a Fully-Connected network, with hidden layers of same size, calling _n_ the number of nodes in the hidden layers, _o_ the output size (10 in permuted, 2 in split), then for each layer:
> * The weights are mapping the activations of the previous layer to the layer in consideration have size _n*n_ (+n if bias is used).
> * The activations, which we pheraps did not define clearly in our previous reply, are the values of each node after the non-linearity and they have size n
> * The feedback matrix maps the error e to the __activations__ so its size is $ o * n $
>
> Secondly, we would like to clarify a key point: DFA updates the __weights__ (which are higly dimensional) by means of the derivative of the error mapped to the __activations__ (which are low dimensional). This is reflected in the term $f'(a_n) h_{n-1}$ in the update equations.
>
> We hope this clarifies your doubt!
>
> The benefit of investigating DFA-diff is to obtain insight into the action of the feedback matrix, and to validate our two hypotheses stated in the beginning of the manuscript. This is __not__ as computationally intensive as defining a new set of weights for every task.

---

### Review · Reviewer_3m68 · 2024-09-15

**Summary Of Contributions:**

The paper addresses catastrophic forgetting by learning a neural network by direct feedback alignment (DFA), compared to back propagation, in both domain incremental and task incremental scenarios of continual learning. Using different DFA strategies, the paper analyzes the effect of DFA for continual learning. For the empirical validations, they use MNIST and FMNIST (two versions of it).

**Audience:**

Yes

**Broader Impact Concerns:**

The proposed work only addresses arguably less challenging continual learning setups instead of various class incremental learning scenarios. Thus the impact of this work is quite limited.

**Claims And Evidence:**

Yes

**Requested Changes:**

- All figures should be in vector graphics for better visibility (currently, all are in bitmap images)
- Comparison to other state of the arts in domain incremental and task incremental
- Empirical study in plain class incremental learning in more realistic settings

**Strengths And Weaknesses:**

**Strengths**

- First work to show DFA is better off at catastrophic forgetting
- Detailed study of using different DFA strategies for CL to compare with

**Weaknesses**

- No theoretical justification why the DFA is better at forgetting than back propagation.
- Continual learning should address complex challenge of balancing between stability (catastrophic forgetting) and plasticity (adaptability to the new task). But this paper lacks of analysis on plasticity except showing overall accuracy (the paper may use 'forward transfer' [A] measure to investigate the plasticity)
- All experiments are with very small scaled datasets. Large scale empirical results will improve the feasibility of the proposed method. Especially, many of the results are in 100% which may not well differentiate the value of the proposed method.
- Insufficient justification of using cosine similarity for the proxy of dataset similarity while CCA and CKA shows similar trend with cosine similarity with different magnitude.
- Domain incremental and task incremental are arguably less challenging setups among class incremental learning scenarios. Given there are many challenging/realistic scenarios proposed for class incremental learning [B,C,D], the empirical validation is not very challenging and is less of importance in the literature.
- While there are a lot of work in continual learning literature, the paper only compare their method to EWC and RF. The reviewer suggests to compare with more state of the arts.

[A] Chen et al., Is forgetting less a good inductive bias for forward transfer?, ICLR 2023\
[B] Prabhu et al., GDumb: A simple approach that questions our progress in continual learning, ECCV 2020\
[C] Bang et al., Online continual learning on a contaminated data stream with blurry task boundaries, CVPR 2022\
[D] Koh et al., Online boundary-free continual learning by scheduled data prior, ICLR 2023

---

> ### Author Response · Authors · 2024-12-02
>
> ## Weaknesses
> > No theoretical justification why the DFA is better at forgetting than back propagation.
>
> Thank you for raising this concern, it gives us a chance to clarify the theoretical _roots_ of our approach.
> Our approach to continual learning is firmly rooted in the theoretical analysis of DFA performed by Refinetti et al. (ICML 2021), who gave a mathematically precise description of the Weight Alignment (WA) and Gradient Alignment (GA) between networks and their feedback matrices, and whcih crucially identified the degeneracy breaking mechanism which is the basis to our approach.
> Our work provides empirical evidence that degeneracy breaking has important ramifications for Continual Larning. A more comprehensive analysis of DFA in continual learning could be carried out by the combination of the existing teacher-student analysis of DFA (Refinetti et al. ICML 2022) and the teacher-student analysis on Continual Learning (Lee et al. ICML 2021, Lee et al. PMLR 2022).
>
>
> > This paper lacks of analysis on plasticity except showing overall accuracy (the paper may use 'forward transfer' [A] measure to investigate the plasticity)
>
> Thank you for the suggestion, we added a discussion about this measure on DFA, see Appendix I: DFA-diff proves once again to train the tasks acting on representations that do not affect the forward transfer of future tasks.
>
> > All experiments are with very small scaled datasets. Large scale empirical results will improve the feasibility of the proposed method. Especially, many of the results are in 100% which may not well differentiate the value of the proposed method.
>
> We added the results on the CIFAR10 dataset, with three colors, to enclose a larger scale results and our claims extend to this dataset. On this benchmark, DFA-same is still the best method in the Domain-IL and DFA-diff is the best method in Task-IL.
>
> > Insufficient justification of using cosine similarity for the proxy of dataset similarity while CCA and CKA shows similar trend with cosine similarity with different magnitude.
>
> Thank you for your comment, clarifying this is indeed useful, we modified the appendix  C to include this considerations.
>
> Cosine similarity measures pairwise angles of the images in the same class among the different datasets and it is sensitive on the “local alignment”. This measure gives a measure of how much the first layer of the network has to be updated among the different tasks because the less the row-inputs are aligned, more the first layer will cause catastrophic forgetting.
>
> This is expecially helpful at inspecting how much the classes share row features, i.e. how much the network will need to put effort in re--learning the features versus how much it can re--use the ones already learned. This is not equivalently captured by CCA or CKA that are tolerant for linear transformations such as translations or orthogonal transformations such as rotations.
>
> ## Requested changes
> > All figures should be in vector graphics for better visibility (currently, all are in bitmap images).
>
> Thank you for the advice, we replaced all the figures with the SVG version.
>
> > Empirical study in plain class incremental learning in more realistic settings
> >Domain incremental and task incremental are arguably less challenging setups among class incremental learning scenarios.
>
> We performed an assessment of DFA's performances in the class incremental scenario and the performances are not as high as the other scenarios because it is indeed a challenging setting. The adaptation of DFA in such realistic settings would require further work, possibly integrating it to already existing strategies. This experiment allowed us to corroborate our claims from an other angle because in DFA-diff the heads that have a non-trivial Accuracy do not decrease while training on other tasks, while this is the case in DFA-same.
>
> > Comparison to other state of the arts in domain incremental and task incremental
> > While there are a lot of work in continual learning literature, the paper only compare their method to EWC and RF. The reviewer suggests to compare with more state of the arts.
>
> We agree that it a baseline like CAB or OWM could support our hypothesis, because they implement the same strategy of orthogonalizing gradients that we hypothesize happens in DFA-diff. Although, these strategies are computationally expensive and there is no available maintained code. We use other tools for supporting our hypothesis: we computed the alignment of the gradients and the overlap of the activations. Other state of the arts comparisons are out of the scope of this project, because they can be used on top of DFA. In fact, DFA can be combined with Memory Replay or Curriculum Learning or other successful strategies.

---

### Decision · Action_Editor_DyHY · 2025-01-20

**Recommendation:** Accept with minor revision

**Comment:**

The reviewers were split about whether the paper should be accepted. The main concern is whether the paper's current contribution will interest the community. A new method (DFA) improving continual learning would be interesting, but does the current manuscript convincingly show this? Does the initial work on this topic even provide enough evidence that DFA is worth further investigation? The lack of theoretical argument combined with the relatively small-scale experiment (even with CIFAR-10) made the reviewers question it.

While some uncertainty remains, I lean in favour of accepting this paper (this was the recommendation of two of the three reviewers). However, **I strongly suggest the authors add a larger-scale experiment (e.g., with Imagenet-1K) as I suspect, based on the reviews, this would more clearly interest the continual learning community.**

Requests:
- In the intro, you write, "We empirically show that DFA performs better at Continual Learning than vanilla back-propagation
and other baselines." Looking at the result, the story seems a bit more nuanced. I suggest updating the contribution to reflect this better.
- The choice of baseline seems to vary (e.g., why was EWC omitted from Table 2?). I suggest uniformizing.
- Formatting. The presentation in the latest version of the paper has improved, but I believe several aspects still require polishing. I think these might negatively affect the perception of the work. I highlight a few here:
   - In the intro, you mention "datasets based on MNIST and FMNIST" (bottom of p. 2), but not the newly introduced CIFAR10
   - I wonder about the abbreviations AF and AP. I don't know if they are standard, and I wonder if using average forgetting/performance everywhere wouldn't improve clarity.
   - In tables, I suggest aligning the numbers across rows
   - Wrong citation style: CIFAR10 Krizhevsky (2009) dataset, and the MNIST dataset Deng (2012). (top of p. 4)
   - Reference to Table 11 jumps to Figure 11 (p. 7)
   - The style of the plots also varies (e.g., the markers aren't always the same and seem to be using different color palettes)


Other suggestions:
- One reviewer suggests the discussion in Appendix A would be helpful in the main paper. I leave it as a suggestion but note that you can go over the initial 12 pages in your camera-ready version.

**Audience:**

On this point, while all three reviewers answered positively (in both their original reviews and final recommendations), there is also a sense that the current investigation might be of limited interest to the community. Again, I make a few suggestions below.

**Claims And Evidence:**

The response period and the latest version of the manuscript answered most of the questions raised by reviewers, and there is consensus that the paper's claims are sufficiently supported. I discuss this below and make a few suggestions and requests.